# A Kernel Perspective on Behavioural Metrics for Markov Decision Processes

**Pablo Samuel Castro**[*]                                  *psc@google.com*
*Google DeepMind*

**Tyler Kastner**[*,†]                          *tyler.kastner@mail.mcgill.ca*
*University of Toronto*

**Prakash Panangaden**[*]                              *prakash@cs.mcgill.ca*
*McGill University*

**Mark Rowland**[*]                                *markrowland@google.com*
*Google DeepMind*

**Reviewed on OpenReview:** *https://openreview.net/forum?id=nHfPXl1ly7*

## Abstract

Behavioural metrics have been shown to be an effective mechanism for constructing representations in reinforcement learning. We present a novel perspective on behavioural metrics for Markov decision processes via the use of positive definite kernels. We leverage this new perspective to define a new metric that is provably equivalent to the recently introduced MICo distance (Castro et al., 2021). The kernel perspective further enables us to provide new theoretical results, which has so far eluded prior work. These include bounding value function differences by means of our metric, and the demonstration that our metric can be *provably* embedded into a finite-dimensional Euclidean space with low distortion error. These are two crucial properties when using behavioural metrics for reinforcement learning representations. We complement our theory with strong empirical results that demonstrate the effectiveness of these methods in practice.

## 1 Introduction

As tabular methods are insufficient for most state spaces, function approximation in reinforcement learning is a well-established paradigm for estimating functions of interest, such as the expected sum of discounted returns $V$ (Sutton & Barto, 2018). A general value function approximator can be seen as a composition of a $m$-dimensional *embedding* $\phi : \mathcal{X} \to \mathbb{R}^m$, which maps a state space $\mathcal{X}$ to a $m$-dimensional space $\mathbb{R}^m$, and a *function approximator* $\psi : \mathbb{R}^m \to \mathbb{R}$ (e.g. $V \approx \psi \circ \phi$). While $\phi$ can be fixed, as is the case in linear function approximation (Baird, 1995; Konidaris et al., 2011) an increasingly popular approach is to learn both $\psi$ and $\phi$ concurrently, typically with the use of deep neural networks. This raises the question of what constitutes a good embedding $\phi$, which forms the basis of the field of representation learning.

---

[*]Authors listed in alphabetical order. See below for details of contributions.
[†]Work done while a student at McGill University.

Previous approaches to learning $\phi$ include spectral decompositions of transition and reward operators (Dayan, 1993; Mahadevan & Maggioni, 2007) and implicit learning through the use of auxiliary tasks (Lange & Riedmiller, 2010; Finn et al., 2015; Jaderberg et al., 2017; Shelhamer et al., 2017; Hafner et al., 2019; Lin et al., 2019; Bellemare et al., 2019; Yarats et al., 2021); such work has shown that learning embeddings plays a crucial role in determining the success of deep reinforcement learning algorithms. This motivates the development of further methods for learning embeddings, and raises the central question as to what properties of an embedding are beneficial in reinforcement learning.

A core role played by the embedding map $\phi$ is to mediate generalisation of value function predictions across states; if the embeddings $\phi(x)$ and $\phi(y)$ of two states $x, y$ are close in $\mathbb{R}^m$, then by virtue of the regularity (or more precisely, Lipschitz continuity) of the function approximator $\psi$, the states $x$ and $y$ will be predicted to have similar values (Gelada et al., 2019; Le Lan et al., 2021). Thus, a good embedding should cluster together states which are known to have similar behavioural properties, whilst keeping dissimilar states apart.

When learning value functions for reinforcement learning, how can one determine whether a representation allows learning generalization between similar states, but avoids *over*-generalization between dissimilar states? Metrics over the environment state space (*state metrics* for short) afford us a natural way to do so via value function upper-bounds, by taking $\phi$ to be a metric embedding of $d$ in $\mathbb{R}^m$. To illustrate the effect of metrics on generalisation, consider two metrics $d_1$ and $d_2$ with the property that $|V(x) - V(y)| \leq d_1(x, y) \ll d_2(x, y)$, where $V : \mathcal{X} \to \mathbb{R}$ is a value function. The metric $d_2$ offers less information about the similarity between $x$ and $y$ than $d_1$ and may suffer from under-generalization[1]. Thus, $d_1$ is a more useful metric for representation learning by virtue of being a *tighter upper-bound*. State metrics satisfying such bounds have accordingly played an important role in representation learning in deep RL (Castro, 2020; Zhang et al., 2021; Castro et al., 2021). Contrastingly, a metric $d$ that violates said upper bound (e.g. for some states $x$ and $y$, $d(x, y) < |V(x) - V(y)|$) may encourage dissimlar states to be mapped near each other, and can thus suffer from *over*-generalization.

Bisimulation metrics (Desharnais et al., 1999; van Breugel & Worrell, 2001; Ferns et al., 2004) are a well established class of state metrics that provide such *value function upper bounds*: $|V(x) - V(y)| \leq d(x, y)$. This translates into a guarantee in the function approximation setting under the assumption that $d$ *can* be embedded into $\mathbb{R}^m$, and that, for example, $\psi$ is 1-Lipschitz continuous, yielding

$$|V(x) - V(y)| \approx |\psi(\phi(x)) - \psi(\phi(y))| \leq \|\phi(x) - \phi(y)\| \approx d(x, y).$$

Bisimulation metrics, however, are expensive to compute, even for tabular systems, due to the fact that one needs to find an optimal coupling between the next-state distributions (Villani, 2008). Castro et al. (2021) overcame this difficulty by replacing the optimal coupling with an independent one, resulting in the MICo distance $U^\pi$, which still satisfies the above-mentioned value function upper bound. Interestingly, this distance is not a proper metric (nor pseudo-metric), but rather a *diffuse metric* (a notion defined by Castro et al. (2021)) that admits non-zero self-distances. The consequences of this are that, when co-learned with $\psi$, the resulting feature map $\phi : \mathcal{X} \to \mathbb{R}^m$ is approximating a *reduced* version of the MICo distance $U^\pi$, denoted as $\Pi U^\pi$. Unfortunately, this reduced MICo does not satisfy the value function upper bound, nor was it demonstrated to be

---

[1] An extreme example of this is to take $d_2(x, y) = \infty \mathbb{1}[x \neq y]$. It is vacuously an upper-bound, but completely uninformative, and will encourage state representations to be mapped far apart from each other.

embeddable in $\mathbb{R}^m$ (the space spanned by neural network representations). Despite demonstrating strong empirical performance, it remained unclear whether the resulting method was theoretically well-founded.

Thus, while there have been empirical successes in using the object $\Pi U^\pi$ as a means of shaping learnt representations in RL, there are two open questions that naturally arise from the preceding discussion:

1. Is it reasonable to assume that $\Pi U^\pi$ is embeddable into a finite-dimensional Euclidean space?

2. Does $\Pi U^\pi$ satisfy some form of continuity, thus potentially explaining its utility in learning representations for RL?

In this work we address these questions directly by taking an alternate view of state similarity: via the use of *kernels*. More precisely, we introduce the concept of a positive-definite kernel on Markov decision processes as a measure of behavioural similarity between states. This new perspective is valuable in itself, as it enables us to leverage reproducing kernel Hilbert space (RKHS) theory for a better understanding of state metrics. We extract a distance from this kernel, and prove its equivalence to the reduced MICo distance. Through this perspective our work provides the following contributions:

- We define a new state distance, $d_{ks}^\pi$ that is constructed from the Hilbert space distance of the RKHS (Section 3.1).

- We demonstrate the equality of this new distance to the reduced MICo $\Pi U^\pi$ of Castro et al. (2021) (Section 3.2).

- We derive a novel result demonstrating that $d_{ks}^\pi$ (and $\Pi U^\pi$ by extension) do serve as an upper bound to value function differences, with an additive component (Section 3.3).

- We prove that $d_{ks}^\pi$ can be embedded into a finite-dimensional Euclidean space with low distortion error (Section 3.4).

- We demonstrate empirically that $d_{ks}^\pi$ performs comparably to MICo when used in a deep reinforcement learning setting (Section 4).

Before doing so, we provide a review of some background material in the next section.

## 2 Background

We provide a brief overview of some of the concepts used throughout this work, and provide a more detailed background in the appendix.

### 2.1 Markov decision processes

We consider a Markov decision process (MDP) given by $(\mathcal{X}, \mathcal{A}, \mathcal{P}, \mathcal{R}, \gamma)$, where $\mathcal{X}$ is a finite state space, $\mathcal{A}$ a set of actions, $\mathcal{P} : \mathcal{X} \times \mathcal{A} \to \mathscr{P}(\mathcal{X})$ a transition kernel, and $\mathcal{R} : \mathcal{X} \times \mathcal{A} \to \mathscr{P}(\mathbb{R})$ a reward kernel ($\mathscr{P}(\mathcal{Z})$ is the set of probability distributions on a measurable set $\mathcal{Z}$). We will write $\mathcal{P}_x^a := \mathcal{P}(x, a)$ for the transition distribution from taking action $a$ in state $x$, $\mathcal{R}_x^a := \mathcal{R}(x, a)$ for

the reward distribution from taking action $a$ in state $x$, and write $r_x^a$ for the expectation of this distribution. A policy $\pi$ is a mapping $\mathcal{X} \to \mathscr{P}(\mathcal{A})$. We use the notation $\mathcal{P}_x^\pi = \sum_{a \in \mathcal{A}} \pi(a|x) \mathcal{P}_x^a$ to indicate the state distribution obtained by following one step of a policy $\pi$ while in state $x$. We use $\mathcal{R}_x^\pi = \sum_{a \in \mathcal{A}} \pi(a|x) \mathcal{R}_x^a$ to represent the reward distribution from $x$ under $\pi$, and $r_x^\pi$ to indicate the expected value of this distribution. Finally, $\gamma \in [0, 1)$ is the discount factor used to compute the discounted long-term return.

The *value* of a policy $\pi$ is the expected total return an agent attains from following $\pi$, and is described by a function $V^\pi : \mathcal{X} \to \mathbb{R}$, such that for each $x \in \mathcal{X}$,

$$V^\pi(x) = \mathbb{E}_\pi \left[ \sum_{t \geq 0} \gamma^t R_t \,\middle|\, X_0 = x \right].$$

Here, $R_t$ is the random variable representing the reward received at time step $t$ when following policy $\pi$ starting at state $x$. An optimal policy $\pi^*$ is a policy which achieves the maximum value function at each state, which we will denote $V^*$. It satisfies the Bellman optimality recurrence:

$$V^*(x) = \max_{a \in \mathcal{A}} \mathbb{E} \left[ R_0 + \gamma V^*(X_1) \,\middle|\, X_0 = x, A_0 = a \right]. \tag{1}$$

## 2.2 The MICo distance

A common approach to dealing with very large, or even infinite, state spaces is via the use of an embedding $\phi : \mathcal{X} \to \mathbb{R}^m$, which maps the original state space into a lower-dimensional representation space. State values can then be learned on top of these representations: e.g. $\hat{V}(x) \approx \psi(\phi(x))$, where $\psi$ is a function approximator.

A desirable property of representations is that they can bound differences with respect to the function of interest (e.g. value functions). Le Lan et al. (2021) argued state behavioural metrics are a useful mechanism for this, as they can often bound differences in values: $d(x, y) \geq |V^*(x) - V^*(y)|$. The structure of the metric impacts its effectiveness: picking $d(x, y) = (R_{\max} - R_{\min})(1 - \gamma)^{-1}$ allows one to satisfy the same upper bound, but in a rather uninformative way.

Bisimulation metrics $d_\sim$ (Ferns et al., 2004) are appealing metrics to use, as they satisfy the above upper bound, while still capturing behavioural similarity that goes beyond simple value equivalence (see Appendix C for a more in-depth discussion of bisimulation metrics). Indeed, various prior works have built reinforcement learning methods based on the notion of bisimulation metrics (Gelada et al., 2019; Agarwal et al., 2021a; Hansen-Estruch et al., 2022). Castro (2020) demonstrated that, with some simplifying assumptions, these metrics are learnable with neural networks; Zhang et al. (2021), Castro et al. (2021), Kemertas & Aumentado-Armstrong (2021) and Kemertas & Jepson (2022) demonstrated they are learnable without the assumptions.

Castro et al. (2021) introduced the *MICo distance*, along with a corresponding differentiable loss, which can be added to any deep reinforcement learning agent without extra parameters. The added loss showed statistically significant performance improvements on the challenging Arcade Learning Environment (Bellemare et al., 2013), as well as on the DeepMind control suite (Tassa et al., 2018).

Our paper builds on the MICo distance, and as such, we present a brief overview of the main definitions and results in this section. We begin with a result that introduces the MICo operator $(T_M^\pi)$, defines the MICo distance as its unique fixed point $(U^\pi)$, and shows this distance provides an upper bound on value differences between two states. Before doing so, we define the *Łukaszyk–Karmowski*

*distance* (LK-distance), which is used in the definition of the MICo distance. A more thorough discussion of the LK-distance is provided in Appendix C.1.2.

**Definition 1.** *Given two probability measures $\mu$ and $\nu$ on a set $\mathcal{X}$ and a metric $d$ on $\mathcal{X}$, the* Łukaszyk–Karmowski distance $d_{ŁK}$ *(Łukaszyk, 2004) is defined as*

$$d_{ŁK}(d)(\mu, \nu) = \int d(x, y) \, \mathrm{d}(\mu \times \nu)(x, y),$$

*or equivalently*

$$d_{ŁK}(d)(\mu, \nu) = \mathop{\mathbb{E}}_{X \sim \mu, Y \sim \nu}[d(X, Y)].$$

**Theorem 2** (Castro et al. (2021)). *Given a policy $\pi$, the MICo operator $T_M^\pi : \mathbb{R}^{\mathcal{X} \times \mathcal{X}} \to \mathbb{R}^{\mathcal{X} \times \mathcal{X}}$, given by $T_M^\pi(U)(x, y) = |r_x^\pi - r_y^\pi| + \gamma \, d_{ŁK}(U)(\mathcal{P}_x^\pi, \mathcal{P}_y^\pi)$, has a unique fixed point $U^\pi$, referred to as the MICo distance.[2] Further, the MICo distance upper bounds the absolute difference between policy-value functions. That is, for $x, y \in \mathcal{X}$, we have $|V^\pi(x) - V^\pi(y)| \le U^\pi(x, y)$.*

The MICo distance was based on the $\pi$-bisimulation metric ($d_\sim^\pi$) introduced by Castro (2020); the key difference is that the update operator for $d_\sim^\pi$ uses the Kantorovich distance between probability measures instead of $d_{ŁK}$ (see Appendix C.1.1 for more details on the Kantorovich metric).

Castro et al. (2021) adapted the MICo distance to be used for learning state feature maps $\phi$. However, the authors demonstrated the features used for control were actually a "reduction" of the diffuse metric approximant; this distance was dubbed the *reduced MICo distance* $\Pi U^\pi$.

**Definition 3** (Reduced MICo). *The reduced MICo distance $\Pi U^\pi$ is defined by*

$$\Pi U^\pi(x, y) = U^\pi(x, y) - \frac{1}{2}(U^\pi(x, x) + U^\pi(y, y)).$$

Two immediate properties of $\Pi U^\pi$ are that it is symmetric, and satisfies $\Pi U^\pi(x, x) = 0$. However, it was unknown whether $\Pi U^\pi$ satisfied the triangle inequality, as well as whether it was positive in general. One negative result shown by Castro et al. (2021) was that the value function upper bound does not hold for $\Pi U^\pi$.

**Proposition 4** (Castro et al. (2021)). *There exists an MDP with $x, y \in \mathcal{X}$, and policy $\pi$ where $|V^\pi(x) - V^\pi(y)| > \Pi U^\pi(x, y)$.*

While Castro et al. (2021) demonstrated the strong empirical performance yielded by the reduced MICo distance in combination with deep reinforcement learning, Proposition 4 highlights that important properties for general state similarity metrics remain unknown for the reduced MICo. We pause here to take stock of what is unknown regarding $\Pi U^\pi$:

- From a purely metric perspective, it is unknown whether $\Pi U^\pi$ is always positive (in general, applying the reduction operator $\Pi$ to a positive function $f$ does not result in $\Pi f$ being positive), or whether $\Pi U^\pi$ satisfies the triangle inequality. These are important to understand so as to guarantee that the learned representations are well-behaved.

- From the behavioural similarity perspective, it is not known whether $\Pi U^\pi$ has any quantitative relationship to the value function $V^\pi$, as Proposition 4 demonstrates the standard value function upper bound does not hold.

---

[2]The distance $d_{ŁK}$ is defined in Appendix C.1.2.

- From a practical perspective, it is unknown $\Pi U^\pi$ is even embeddable in $\mathbb{R}^m$.

In this work we seek to resolve these mysteries through the lens of reproducing kernel Hilbert spaces, which we introduce next (a more extensive discussion is provided in Appendix C.5).

### 2.3 Reproducing kernel Hilbert spaces

Let $\mathcal{X}$ be a finite [3] set, and define a function $k : \mathcal{X} \times \mathcal{X} \to \mathbb{R}$ to be a positive definite kernel if it is symmetric and positive definite:[4] for any $\{x_1, \ldots, x_n\} \in \mathcal{X}$, $\{c_1, \ldots, c_n\} \in \mathbb{R}$, we have that

$$\sum_{i=1}^{n} \sum_{j=1}^{n} c_i c_j k(x_i, x_j) \geq 0.$$

We will often use *kernel* as a shorthand for positive definite kernel. Given a kernel $k$ on $\mathcal{X}$, one can construct a space of functions $\mathcal{H}_k$ referred to as a reproducing kernel Hilbert space (RKHS), through the following steps:[5]

(i) Construct a vector space of real-valued functions on $\mathcal{X}$ of the form $\{k(x, \cdot) : x \in \mathcal{X}\}$.

(ii) Equip this space with an inner product given by $\langle k(x, \cdot), k(y, \cdot) \rangle_{\mathcal{H}_k} = k(x, y)$.

(iii) Take the completion of the vector space with respect to the inner product $\langle \cdot, \cdot \rangle_{\mathcal{H}_k}$.

The Hilbert space obtained at the end of step (iii) is the reproducing kernel Hilbert space for $k$.

It is common to introduce the notation $\varphi(x) := k(x, \cdot)$, where $\varphi : \mathcal{X} \to \mathcal{H}$ is often called the *feature map*, and $\varphi(x)$ is understood as the *embedding* of $x$ in $\mathcal{H}$. Note that we are using $\varphi$ to represent the mapping of states onto a Hilbert space (e.g. $\varphi : \mathcal{X} \to \mathcal{H}_k$), which is distinct from the symbol $\phi$ which we use to represent the mapping of states onto a Euclidean space (e.g. $\phi : \mathcal{X} \to \mathbb{R}^m$). One can also embed probability distributions on $\mathcal{X}$ in $\mathcal{H}_k$. Given a probability distribution $\mu$ on $\mathcal{X}$, one can define the embedding of $\mu$, $\Phi(\mu) \in \mathcal{H}_k$ as

$$\Phi(\mu) = \mathbb{E}_{X \sim \mu}[\varphi(X)] = \int_{\mathcal{X}} \varphi(x) d\mu(x),$$

where the integral taken is a Bochner integral,[6] as we are integrating over $\mathcal{H}_k$-valued functions. The embeddings of measures into $\mathcal{H}_k$ allow one to easily compute integrals, as one can show using the Riesz representation theorem that for $f \in \mathcal{H}_k$, one has

$$\int_{\mathcal{X}} f d\mu = \langle f, \Phi(\mu) \rangle_{\mathcal{H}_k}.$$

These embeddings also allow us to define metrics on $\mathcal{X}$ and $\mathscr{P}(\mathcal{X})$ by looking at the Hilbert space distance of their embeddings.

---

[3] The general theory of reproducing kernel Hilbert spaces holds for much more general classes of sets (Aronszajn, 1950), but as mentioned earlier in the paper, our work focuses on MDPs with finite state spaces, so we present RKHS theory in the context of finite sets, allowing us to avoid some mathematical technicalities.

[4] We remark that the definition of positive definite is not consistent across the literature. We follow the convention of the kernel methods community, and define a function to be strictly positive definite if the inequality is strict unless $c_1 = \cdots = c_n = 0$. In the linear algebra and optimization communities however, this is referred to as positive definite, and the definition provided is referred to as positive semidefinite.

[5] An RKHS can be alternately defined by choosing a suitable set of functions and constructing the kernel $k$ from this set, we review this approach in Appendix C.5.2.

[6] A generalization of the Lebesgue integral to functions taking values in a Banach space, further details can be found in Arendt et al. (2001).

**Definition 5.** *Given a positive definite kernel $k$, define $\rho_k$ as its induced distance:*

$$\rho_k(x,y) := \|\varphi(x) - \varphi(y)\|_{\mathcal{H}_k}.$$

*By expanding the inner product, the squared distance can be written solely in terms of the kernel $k$:*

$$\rho_k^2(x,y) = k(x,x) + k(y,y) - 2k(x,y).$$

We can perform the same process to construct a metric on $\mathscr{P}(\mathcal{X})$ using $\Phi$:

**Definition 6** (Gretton et al. (2012)). *Let $k$ be a kernel on $\mathcal{X}$, and $\Phi : \mathscr{P}(\mathcal{X}) \to \mathcal{H}_k$ be as defined above. Then the Maximum Mean Discrepancy (MMD) is a pseudometric on $\mathscr{P}(\mathcal{X})$ defined by*

$$MMD(k)(\mu,\nu) = \|\Phi(\mu) - \Phi(\nu)\|_{\mathcal{H}_k}.$$

A *semimetric* is a distance function which respects all metric axioms save for the triangle inequality. A semimetric space $(\mathcal{X}, \rho)$ is of *negative type* if for all $x_1, \ldots, x_n \in \mathcal{X}$, $c_1, \ldots, c_n \in \mathbb{R}$ such that $\sum_{i=1}^n c_i = 0$, we have

$$\sum_{i=1}^n \sum_{j=1}^n c_i c_j \rho(x_i, x_j) \leq 0.$$

Given a semimetric of negative type $\rho$ on $\mathcal{X}$, we can define a distance on $\mathscr{P}(\mathcal{X})$ known as the *energy distance*, defined as

$$\mathcal{E}(\rho)(\mu,\nu) = \mathop{\mathbb{E}}_{X \sim \mu, Y \sim \nu}[\rho(X,Y)] - \frac{1}{2}\left(\mathop{\mathbb{E}}_{X_1,X_2 \sim \mu}[\rho(X_1,X_2)] + \mathop{\mathbb{E}}_{Y_1,Y_2 \sim \nu}[\rho(Y_1,Y_2)]\right),$$

where the pairs of random variables in each expectation are independent.

If $\rho$ is of negative type, this guarantees that we have $\mathcal{E}(\rho)(\mu,\nu) \geq 0$ for all $\mu, \nu \in \mathscr{P}(\mathcal{X})$. Semimetrics of negative type have a connection to positive definite kernels, as shown in Sejdinovic et al. (2013): the induced distance squared $\rho_k^2$ is a semimetric of negative type, which we say is induced by $k$. Conversely, a semimetric of negative type $\rho$ induces a family of positive definite kernels $K_\rho$ parametrised by a chosen base point $x_0 \in \mathcal{X}$:

$$K_\rho^{x_0}(x,x') = \frac{1}{2}(\rho(x,x_0) + \rho(x',x_0) - \rho(x,x')).$$

The relationship is symmetric, so that each kernel $k \in K_\rho$ has $\rho$ as its induced semimetric. With this symmetry in mind, we call a kernel $k$ and a semimetric of negative type an *equivalent pair* if they induce one another through the above construction. This equivalence does not only live in $\mathcal{X}$ however, as the following proposition shows that it lifts into $\mathscr{P}(\mathcal{X})$ as well.

**Proposition 7** (Sejdinovic et al. (2013)). *Let $(k, \rho)$ be an equivalent pair, and let $\mu, \nu \in \mathscr{P}(\mathcal{X})$. Then we have the equivalence*

$$MMD^2(k)(\mu,\nu) = \mathcal{E}(\rho)(\mu,\nu).$$

The central takeaway of the equivalences proved in Sejdinovic et al. (2013) is that using metrics of negative type and positive definite kernels are two perspectives of the same underlying structure.

### 2.4 Prior work using RKHS for MDPs

We are not the first to consider using the theory of reproducing kernel Hilbert spaces in the context of Markov decision processes. Indeed, Ormoneit & Sen (2002) proposed to use a fixed kernel to perform approximate dynamic programming in RL problems with Euclidean state spaces; a variety of theoretical and empirical extensions of this approach have since been considered (Lever et al., 2016; Barreto et al., 2016; Domingues et al., 2021). The use of fixed RKHSs for approximate dynamic programming in more general state spaces has been studied by Farahmand et al. (2016), Yang et al. (2020), and Koppel et al. (2021). In contrast, as we will discuss below, our work *learns* a kernel on the state space, which classifies states as similar based on their behavioural similarity in the MDP.

## 3 Kernel similarity metrics

We take a new perspective on behavioural metrics in MDPs, through the use of positive definite kernels. We define a contractive operator on the space of kernels, and show that its unique fixed point induces a behavioural distance in a reproducing kernel Hilbert space, and then prove that this distance coincides with the reduced MICo distance $\Pi U^\pi$. We then present new properties of $\Pi U^\pi$ obtained through this perspective.

### 3.1 Definition

Given an MDP $M = (\mathcal{X}, \mathcal{A}, \mathcal{P}, \mathcal{R}, \gamma)$, state similarity metrics which are variants of bisimulation generally follow the form

$$d(x, y) = d_1(x, y) + \gamma \, d_2(d)(\mathcal{P}(x), \mathcal{P}(y)),$$

for states $x, y$ in $\mathcal{X}$. Here, $d_1$ is a distance on $\mathcal{X}$ representing *one-step differences* between $x$ and $y$ (e.g. reward difference), and $d_2$ "lifts" a distance on $\mathcal{X}$ onto a distance on $\mathscr{P}(\mathcal{X})$; thus, $d_2(d)(\mathcal{P}(x), \mathcal{P}(y))$ represents the *long-term behavioural distance* between $x$ and $y$. It is worth noting the similarity to the Bellman optimality recurrence introduced in Equation 1.

In this section we take a similar approach, except rather than quantifying the *difference* of states (metrics), we consider quantifying the *similarity* of states (positive definite kernels). Following this idea, we can define a state similarity kernel $k : \mathcal{X} \times \mathcal{X} \to \mathbb{R}$ as a positive definite kernel which takes the following form:

$$k(x, y) = k_1(x, y) + \gamma \, k_2(k)(\mathcal{P}(x), \mathcal{P}(y)).$$

Similarly to the above expression for $d$, $k_1$ is a kernel on $\mathcal{X}$ which measures the *immediate similarity* of two states $x$ and $y$, and $k_2$ lifts a kernel on $\mathcal{X}$ into a kernel on $\mathscr{P}(\mathcal{X})$.

We can now present a candidate state similarity kernel. We follow Castro (2020) and Castro et al. (2021) in measuring behavioural similarity under a fixed policy, in contrast to measuring behaviour across all possible actions, as done in bisimulation (Ferns et al., 2004; 2011). Following this, we fix a policy $\pi$ which will be the policy under which we measure similarity. For the immediate similarity kernel, we will assume that $\text{supp}(\mathcal{R}) \subseteq [-1, 1]$,[7] and we set $k_1(x, y) = 1 - \frac{1}{2}|r_x^\pi - r_y^\pi|$, which lies in $[0, 1]$. This is a reasonable measure of immediate similarity, as it is maximised when two states have identical immediate rewards, and minimised when two states have maximally distant immediate

---

[7]This assumption is purely for the clarity of presentation, and can be relaxed to assuming boundedness of reward and setting $k_1(x, y) = 1 - \frac{1}{\mathcal{R}_{\max} - \mathcal{R}_{\min}}|r_x^\pi - r_y^\pi|$.

rewards. To lift a kernel $k$ into a kernel on $\mathscr{P}(\mathcal{X})$, we can use the kernel lifting construction given in Guilbart (1979), and define $k_2(k)(\mu, \nu) = \mathbb{E}_{X \sim \mu, Y \sim \nu}[k(X, Y)]$. Combining these, we can define an operator on the space of kernels whose fixed point would be our kernel of interest.

**Definition 8.** *Let $\mathscr{K}(\mathcal{X})$ be the space of positive definite kernels on $\mathcal{X}$. Given $\pi \in \mathscr{P}(\mathcal{A})^{\mathcal{X}}$, the kernel similarity operator $T_k^\pi : \mathscr{K}(\mathcal{X}) \to \mathscr{K}(\mathcal{X})$ is*

$$T_k^\pi(k)(x, y) = \left(1 - \frac{1}{2}|r_x^\pi - r_y^\pi|\right) + \gamma \mathop{\mathbb{E}}_{X' \sim \mathcal{P}_x^\pi, Y' \sim \mathcal{P}_y^\pi}[k(X', Y')].$$

The fact that $T_k^\pi$ indeed maps $\mathscr{K}(\mathcal{X})$ to $\mathscr{K}(\mathcal{X})$ follows from the previous paragraph describing that each operator is a kernel, and that the sum of two kernels is a kernel (Aronszajn, 1950). We now present two lemmas which are necessary to conclude whether a unique fixed point of $T_k^\pi$ exists.

**Lemma 9.** *$T_k^\pi$ is a contraction with modulus $\gamma$ in $\|\cdot\|_\infty$.*

*Proof.* Let $k_1, k_2 \in \mathscr{K}(\mathcal{X})$, we can then write out

$$\|T_k^\pi(k_1) - T_k^\pi(k_2)\|_\infty = \max_{(x,y) \in \mathcal{X} \times \mathcal{X}} |T_k^\pi(k_1)(x, y) - T_k^\pi(k_2)(x, y)|$$

$$= \gamma \max_{(x,y) \in \mathcal{X} \times \mathcal{X}} \left| \mathop{\mathbb{E}}_{X' \sim \mathcal{P}_x^\pi, Y' \sim \mathcal{P}_y^\pi}[k_1(X', Y')] - \mathop{\mathbb{E}}_{X' \sim \mathcal{P}_x^\pi, Y' \sim \mathcal{P}_y^\pi}[k_2(X', Y')] \right|$$

$$= \gamma \max_{(x,y) \in \mathcal{X} \times \mathcal{X}} \left| \mathop{\mathbb{E}}_{X' \sim \mathcal{P}_x^\pi, Y' \sim \mathcal{P}_y^\pi}[k_1(X', Y') - k_2(X', Y')] \right|$$

$$\leq \gamma \|k_1 - k_2\|_\infty. \qquad \square$$

**Lemma 10.** *The metric space $(\mathscr{K}(\mathcal{X}), \|\cdot\|_\infty)$ is complete.*

*Proof.* As we assume $\mathcal{X}$ is finite, the space of functions $\mathbb{R}^{\mathcal{X} \times \mathcal{X}}$ is a finite-dimensional Euclidean vector space, and hence is complete with respect to the $L^\infty$ norm. It therefore suffices to show that $\mathscr{K}(\mathcal{X})$ is closed in $\mathbb{R}^{\mathcal{X} \times \mathcal{X}}$. We can consider a sequence $\{k_n\}_{n \geq 0}$ in $\mathscr{K}(\mathcal{X})$ which converges to $k \in \mathbb{R}^{\mathcal{X} \times \mathcal{X}}$ in $\|\cdot\|_\infty$ and show that $k \in \mathscr{K}(\mathcal{X})$. This is equivalent to showing that $k$ is both symmetric and positive definite, which follows immediately from the fact that each $k_n$ is and the convergence is uniform. Hence $\mathscr{K}(\mathcal{X})$ is closed with respect to $\|\cdot\|_\infty$, and thus is complete. $\qquad \square$

With these two lemmas, we can now show that the required fixed point indeed exists.

**Proposition 11.** *There is a unique kernel $k^\pi$ satisfying*

$$k^\pi(x, y) = \left(1 - \frac{1}{2}|r_x^\pi - r_y^\pi|\right) + \gamma \mathop{\mathbb{E}}_{X' \sim \mathcal{P}_x^\pi, Y' \sim \mathcal{P}_y^\pi}[k^\pi(X', Y')].$$

*Proof.* Combining Lemma 9 and Lemma 10, we know that the operator $T_k^\pi$ is a contraction in a complete metric space. We can now use Banach's fixed point theorem to obtain the existence of a unique fixed point, which is $k^\pi$. $\qquad \square$

Having a kernel on our MDP now gives us an RKHS of functions on the MDP, which we refer to as $\mathcal{H}_{k^\pi}$. Moreover, we have an embedding of each state into $\mathcal{H}_{k^\pi}$ given by $\varphi^\pi(x) = k^\pi(x, \cdot)$. Using this construction, we can define a distance between states in $\mathcal{X}$ by considering their Hilbert space distance in $\mathcal{H}_{k^\pi}$.

**Definition 12.** *We define the **kernel similarity metric (KSMe)** as the distance function*

$$d_{ks}^\pi(x, y) := \|\varphi^\pi(x) - \varphi^\pi(y)\|_{\mathcal{H}_{k^\pi}}^2.$$

## 3.2 Equivalence with reduced MICo distance

We will prove a number of useful theoretical properties of $d_{ks}^\pi$ in the rest of this section. But first, we demonstrate that $d_{ks}^\pi$ is equal to the reduced MICo ($\Pi U^\pi$) from Castro et al. (2021). Given that Castro et al. (2021) left a number of unresolved properties of $\Pi U^\pi$, this equality will be important for the remainder of the paper as it means the new theoretical insights we prove for $d_{ks}^\pi$ also hold for $\Pi U^\pi$.

Referring to Section 2.3, we have that $d_{ks}^\pi$ is the semimetric of negative type induced by $k^\pi$. We now demonstrate that $d_{ks}^\pi$ can be written as a sum of reward distance and transition distribution distance, similar to the form of the behavioural metrics discussed in Section 3.1.

**Proposition 13.** *The kernel similarity metric $d_{ks}^\pi$ satisfies*

$$d_{ks}^\pi(x,y) = |r_x^\pi - r_y^\pi| + \gamma MMD^2(k^\pi)(\mathcal{P}_x^\pi, \mathcal{P}_y^\pi).$$

*Proof.* To see this, we can write out the squared Hilbert space distance

$$
\begin{aligned}
d_{ks}^\pi(x,y) &= \|\varphi^\pi(x) - \varphi^\pi(y)\|_{\mathcal{H}_{k^\pi}}^2 \\
&= k^\pi(x,x) + k^\pi(y,y) - 2k^\pi(x,y) \\
&= |r_x^\pi - r_y^\pi| + \gamma\langle\Phi(\mathcal{P}_x^\pi),\Phi(\mathcal{P}_x^\pi)\rangle_{\mathcal{H}_{k^\pi}} + \gamma\langle\Phi(\mathcal{P}_y^\pi),\Phi(\mathcal{P}_y^\pi)\rangle_{\mathcal{H}_{k^\pi}} - 2\gamma\langle\Phi(\mathcal{P}_x^\pi),\Phi(\mathcal{P}_y^\pi)\rangle_{\mathcal{H}_{k^\pi}} \\
&= |r_x^\pi - r_y^\pi| + \gamma MMD^2(k^\pi)(\mathcal{P}_x^\pi,\mathcal{P}_y^\pi),
\end{aligned}
$$

where in the first equality we expanded the norm as in Definition 5, and in the second line we used

$$
\begin{aligned}
k^\pi(x,x) &= \gamma \mathop{\mathbb{E}}_{X_1',X_2'\sim\mathcal{P}_x^\pi}[k(X_1',X_2')] \\
&= \gamma\langle\Phi(\mathcal{P}_x^\pi),\Phi(\mathcal{P}_x^\pi)\rangle_{\mathcal{H}_{k^\pi}}. \qquad \square
\end{aligned}
$$

Before presenting the main result of this section (Theorem 15), we state a necessary technical lemma. The proof is provided in Appendix A.

**Lemma 14.** *For any measures $\mu$, $\nu$, and $n \geq 0$, we have that*

$$\mathop{\mathbb{E}}_{\substack{X_1,X_2\sim\mu \\ Y_1,Y_2\sim\nu}}[k_n(X_1,X_2) + k_n(Y_1,Y_2) - 2k_n(X_1,Y_1)] = \mathop{\mathbb{E}}_{\substack{X_1,X_2\sim\mu \\ Y_1,Y_2\sim\nu}}\left[U_n(X_1,Y_1) - \frac{1}{2}(U_n(X_1,X_2) + U_n(Y_1,Y_2))\right].$$

The next result proves the equivalence of $d_{ks}^\pi$ and $\Pi U^\pi$; the implications of this result are that *all of the properties proved for $d_{ks}^\pi$ also apply to $\Pi U^\pi$*. This is significant, as it implies $\Pi U^\pi$ *can* be incorporated into a non-vacuous upper bound of value function differences (Theorem 16), and is guaranteed to be embeddable in a finite-dimensional Euclidean space of appropriate dimension (Theorem 19), addressing the two core theoretical questions raised in the introduction of this paper.

**Theorem 15.** *For any $x,y \in \mathcal{X}$, we have that $d_{ks}^\pi(x,y) = \Pi U^\pi(x,y)$.*

*Proof.* To begin, we will make use of the sequences $(k_n)_{n\geq 0}$, $(U_n)_{n\geq 0}$ defined by $k_n \equiv 0$, $k_{n+1} = T_k^\pi(k_n)$, $U_n \equiv 0$, $U_{n+1} = T_M^\pi(U_n)$. Since both $T_k^\pi$ and $T_M^\pi$ are contractions, we know that $k_n \to k^\pi$ and $U_n \to U^\pi$ uniformly. To prove the statement, we will show that for all $n \geq 0$ and $x,y \in \mathcal{X}$, we have that

$$k_n(x,x) + k_n(y,y) - 2k_n(x,y) = U_n(x,y) - \frac{1}{2}(U_n(x,x) + U_n(y,y)).$$

We can write out

$$k_n(x,x) + k_n(y,y) - 2k_n(x,y)$$
$$= |r_x^\pi - r_y^\pi| + \gamma \mathop{\mathbb{E}}_{\substack{X_1,X_2\sim\mathcal{P}_x^\pi \\ Y_1,Y_2\sim\mathcal{P}_y^\pi}} [k_n(X_1,X_2) + k_n(Y_1,Y_2) - 2k_n(X_1,Y_1)]$$
$$= |r_x^\pi - r_y^\pi| + \gamma \mathop{\mathbb{E}}_{\substack{X_1,X_2\sim\mathcal{P}_x^\pi \\ Y_1,Y_2\sim\mathcal{P}_y^\pi}} \left[ U_n(X_1,Y_1) - \frac{1}{2}(U_n(X_1,X_2) + U_n(Y_1,Y_2)) \right] (\star)$$
$$= U_n(x,y) - \frac{1}{2}(U_n(x,x) + U_n(y,y)),$$

where $(\star)$ follows from Lemma 14. Since $(k_n)_{n\geq 0}$ and $(U_n)_{n\geq 0}$ both converge uniformly, we can take limits and conclude that

$$d_{ks}^\pi(x,y) = k^\pi(x,x) + k^\pi(y,y) - 2k^\pi(x,y) = U^\pi(x,y) - \frac{1}{2}(U^\pi(x,x) + U^\pi(y,y)) = \Pi U^\pi(x,y).$$

$\square$

### 3.3 An additive value function upper bound

Proposition 4 asserts that $\Pi U^\pi$, and hence $d_{ks}^\pi$, does not upper bound the absolute difference in value functions. However, the kernel perspective allows us to show that it satisfies an upper bound with an additive constant. For $x \in \mathcal{X}$ we introduce the notation

$$\Delta_n^\pi(x) = \mathop{\mathbb{E}}_{X'\sim(\mathcal{P}_x^\pi)^n} \left[ \mathop{\mathbb{E}}_{X_1'',X_2''\sim\mathcal{P}_{X'}^\pi} \left[ |r_{X_1''}^\pi - r_{X_2''}^\pi| \right] \right].$$

Intuitively, $\Delta_n^\pi(x)$ is the expected absolute reward difference in two trajectories from $x$, where the trajectories are coupled for the first $n$ steps, and proceed independently for the final $(n+1)$th step. With this quantity, we present the following theorem.

**Theorem 16.** *For any $x,y \in \mathcal{X}$, we have*

$$|V^\pi(x) - V^\pi(y)| \leq \Pi U^\pi(x,y) + \frac{1}{2}\sum_{n\geq 0} \gamma^n(\Delta_n^\pi(x) + \Delta_n^\pi(y)).$$

*Proof.* To begin, we will make use of the sequences $(k_m)_{m\geq 0}$, $(V_m)_{m\geq 0}$ defined by $k_m \equiv 0$, $k_{m+1} = T_k^\pi(k_m)$, $V_m \equiv 0$, $V_{m+1} = T^\pi(V_m)$. Since $T_k^\pi$ and $T^\pi$ are both contractions, we know that $k_m \to k^\pi$ and $V_m \to V^\pi$ uniformly. We will refer to the semimetric equivalent to the $m$th kernel iterate as $d_m$: $d_m(x,y) = k_m(x,x) + k_m(y,y) - 2k_m(x,y)$. We will now use induction to prove that for all $m$, we have that

$$|V_m(x) - V_m(y)| \leq d_m(x,y) + \frac{1}{2}\sum_{n=0}^{m} \gamma^n(\Delta_n^\pi(x) + \Delta_n^\pi(y)).$$

The base case $m = 0$ is immediate, as the left hand side is identically 0. We can now assume the induction hypothesis, and write out

$$|V_{m+1}(x) - V_{m+1}(y)|$$
$$= \left| r_x^\pi + \gamma \mathop{\mathbb{E}}_{X'\sim\mathcal{P}_x^\pi}[V_m(x)] - \left( r_y^\pi + \gamma \mathop{\mathbb{E}}_{Y'\sim\mathcal{P}_y^\pi}[V_m(Y')] \right) \right|$$

$$\leq |r_x^\pi - r_y^\pi| + \gamma \mathop{\mathbb{E}}_{X'\sim\mathcal{P}_x^\pi, Y'\sim\mathcal{P}_y^\pi}\left[\,|V_m(X') - V_m(Y')|\,\right]$$

$$\leq |r_x^\pi - r_y^\pi| + \gamma \mathop{\mathbb{E}}_{X'\sim\mathcal{P}_x^\pi, Y'\sim\mathcal{P}_y^\pi}\left[d_m(X', Y') + \frac{1}{2}\sum_{n=0}^{m}\gamma^n(\Delta_n^\pi(X') + \Delta_n^\pi(Y'))\right]$$

$$= |r_x^\pi - r_y^\pi| + \gamma \mathop{\mathbb{E}}_{X'\sim\mathcal{P}_x^\pi, Y'\sim\mathcal{P}_y^\pi}\left[d_m(X', Y')\right] + \frac{1}{2}\sum_{n=1}^{m+1}\gamma^n(\Delta_n^\pi(x) + \Delta_n^\pi(y))$$

$$= |r_x^\pi - r_y^\pi| + \gamma \mathop{\mathbb{E}}_{X'\sim\mathcal{P}_x^\pi, Y'\sim\mathcal{P}_y^\pi}\left[d_m(X', Y')\right] + \frac{1}{2}\mathop{\mathbb{E}}_{\substack{X',X''\sim\mathcal{P}_x^\pi\\Y',Y''\sim\mathcal{P}_y^\pi}}\left[|r_{X'}^\pi - r_{X''}^\pi| + |r_{Y'}^\pi - r_{Y''}^\pi|\right]$$

$$\qquad + \frac{1}{2}\sum_{n=1}^{m+1}\gamma^n(\Delta_n^\pi(x) + \Delta_n^\pi(y))$$

$$= |r_x^\pi - r_y^\pi| + \gamma \mathop{\mathbb{E}}_{X'\sim\mathcal{P}_x^\pi, Y'\sim\mathcal{P}_y^\pi}\left[d_m(X', Y')\right] + \frac{1}{2}\sum_{n=0}^{m+1}\gamma^n(\Delta_n^\pi(x) + \Delta_n^\pi(y))$$

$$= d_{m+1}(x, y) + \frac{1}{2}\sum_{n=0}^{m+1}\gamma^n(\Delta_n^\pi(x) + \Delta_n^\pi(y)),$$

where we used $\mathbb{E}_{X'\sim\mathcal{P}_x^\pi}[\Delta_n^\pi(X')] = \Delta_{n+1}^\pi(x)$. We note that the sum $\frac{1}{2}\sum_{n=0}^{\infty}\gamma^n(\Delta_n^\pi(x) + \Delta_n^\pi(y))$ is almost surely finite, as $\Delta_n^\pi(x) \leq 1$ almost surely (since we assume $\mathrm{supp}(\mathcal{R}) \subseteq [-1, 1]$), so that

$$\sum_{n=0}^{\infty}\frac{\gamma^n}{2}\left(\Delta_n^\pi(x) + \Delta_n^\pi(y)\right) \leq \sum_{n=0}^{\infty}\gamma^n = \frac{1}{1-\gamma}. \qquad \square$$

With this theorem, it is apparent that the amount by which the bound is broken is controlled by the amount of dispersion in reward coming from the transition probability function $\mathcal{P}^\pi$. A standard tool to measure the dispersion of a measure on $\mathbb{R}$ is the variance, which is not directly applicable in this setting as $\mathcal{P}^\pi$ maps to measures on $\mathcal{X}$. However, we can map each state $x \in \mathcal{X}$ to a real value $r_x^\pi$, and we use this to apply the variance. With this motivation, we define the reward variance of $\mathcal{P}^\pi$ for $x \in \mathcal{X}$ as

$$\mathrm{Var}_\mathcal{R}(\mathcal{P}_x^\pi) = \mathop{\mathbb{E}}_{X'\sim\mathcal{P}_x^\pi}\left[(r_{X'}^\pi)^2\right] - \left(\mathop{\mathbb{E}}_{X'\sim\mathcal{P}_x^\pi}[r_{X'}^\pi]\right)^2.$$

With this definition in mind, we can now demonstrate that the maximal reward variance of an MDP controls the amount the value function difference upper bound is violated.

**Proposition 17.** *Suppose there exists $\sigma^2 \in \mathbb{R}$ such that for each $x \in \mathcal{X}$, $\mathrm{Var}_\mathcal{R}(\mathcal{P}_x^\pi) \leq \sigma^2$. Then for every $x \in \mathcal{X}$ and $k \geq 0$ we have that $\Delta_k^\pi(x) \leq \sqrt{2}\sigma$, and in particular*

$$\left|V^\pi(x) - V^\pi(y)\right| \leq \Pi U^\pi(x, y) + \frac{\sqrt{2}\sigma}{1-\gamma}.$$

*Proof.* We first note that it suffices to show that for any $x$ we have that $\Delta_1^\pi(x) \leq \sqrt{2}\sigma$, since for any $n > 1$ we have $\Delta_n^\pi(x) = \mathbb{E}_{X'\sim(\mathcal{P}_x^\pi)^{n-1}}[\Delta_1^\pi(X')]$:

$$\mathop{\mathbb{E}}_{X'\sim(\mathcal{P}_x^\pi)^{n-1}}[\Delta_1^\pi(X_1)] = \mathop{\mathbb{E}}_{X_{n-1}\sim(\mathcal{P}_x^\pi)^{n-1}}\left[\mathop{\mathbb{E}}_{X_n\sim\mathcal{P}_{X_{n-1}}^\pi}\left[\mathop{\mathbb{E}}_{X_{(n+1)a}, X_{(n+1)b}\sim\mathcal{P}_{X_n}^\pi}\left[|r_{X_{(n+1)a}}^\pi - r_{X_{(n+1)b}}^\pi|\right]\right]\right]$$

$$= \mathop{\mathbb{E}}_{X_n \sim (\mathcal{P}_x^\pi)^n} \left[ \mathop{\mathbb{E}}_{X_{(n+1)a}, X_{(n+1)b} \sim \mathcal{P}_{X_n}^\pi} \left[ |r_{X_{(n+1)a}}^\pi - r_{X_{(n+1)b}}^\pi| \right] \right]$$
$$= \Delta_n^\pi(x),$$

where for clarity we used the notation $X_m$ to denote a random state after taking $m$ steps in the trajectory.

We can first recall an equivalent formula for variance as $\mathrm{Var}(\mu) = \frac{1}{2} \mathbb{E}_{X,Y \sim \mu}[|X - Y|^2]$. Using this and Jensen's inequality, we have that for any $x \in \mathcal{X}$,

$$2\sigma^2 \geq \mathop{\mathbb{E}}_{X' \sim \mathcal{P}_x^\pi, Y' \sim \mathcal{P}_y^\pi}[|r_{X'}^\pi - r_{Y'}^\pi|^2]$$

$$\geq \left( \mathop{\mathbb{E}}_{X' \sim \mathcal{P}_x^\pi, Y' \sim \mathcal{P}_y^\pi}[|r_{X'}^\pi - r_{Y'}^\pi|] \right)^2.$$

Taking the square root of both sides we obtain that $\Delta_1^\pi(x) \leq \sqrt{2}\sigma$, which combined with the above completes the proof. $\qquad\square$

## 3.4 Distortion error bounds on Euclidean embeddings

While we consider various distance spaces $(\mathcal{X}, d)$ on the *ground states* $\mathcal{X}$ of the MDP, in practice we typically work with a representation $(\phi(x) : x \in \mathcal{X}) \in (\mathbb{R}^m)^{\mathcal{X}}$ of the state space, from which values can subsequently be predicted with neural network function approximation. This highlights an important issue in moving from the study of behavioural metrics as abstract mathematical objects to tools for shaping neural representations, which has received relatively little attention so far. Namely, does there exist an embedding $\phi : \mathcal{X} \to \mathbb{R}^m$ such that $\|\phi(x) - \phi(y)\| = d(x, y)$ for all $x, y \in \mathcal{X}$? Stated more concisely, we may ask whether the metric space $(\mathcal{X}, d)$ *embeds into* the Euclidean space $\mathbb{R}^m$; this is an instance of the core problem of study in the field of metric embedding theory (Deza & Laurent, 1997; Matousek, 2013), and there are several central results from this field that can be employed to cast light on the embeddability of behavioural metrics.

As an initial note of caution, Schoenberg (1935) gives a precise characterisation of which finite metric spaces can be embedded into Euclidean space, a consequence of which is that many finite metric spaces cannot be embedded into Euclidean spaces of *any* dimension. A potential upshot is that attempting to learn exact embeddings of behavioural metrics may not be possible, even in small-scale settings. However, the kernel perspective taken earlier in the paper allows us to make immediate progress on the question of embeddability in the specific case of the reduced MICo metric. In particular, since Theorem 15 establishes that the reduced MICo metric can be embedded into the Hilbert space $\mathcal{H}_{k^\pi}$, we can deduce the following result.

**Corollary 18.** *The reduced MICo metric $\Pi U^\pi$ can be embedded into the space $\mathbb{R}^{|\mathcal{X}|}$ with squared Euclidean metric.*

*Proof.* From Theorem 15 and Definition 12, we have that $\Pi U^\pi(x, y) = \|\varphi^\pi(x) - \varphi^\pi(y)\|_{\mathcal{H}_{k^\pi}}^2$ for all $x, y \in \mathcal{X}$. Since $\mathcal{H}_{k^\pi}$ is a Hilbert space of dimension at most $\mathcal{X}$, there is an isometry $\psi : \mathcal{H}_{k^\pi} \to \mathbb{R}^k$ for some $k \leq |\mathcal{X}|$. The composition $\psi \circ \varphi^\pi$ therefore embeds $\Pi U^\pi$ exactly in $\mathbb{R}^k$ under the squared Euclidean metric. $\qquad\square$

In many practical settings, the dimensionality of the space $\mathbb{R}^m$ into which the representation function $\varphi$ maps is generally taken to be much smaller than $|\mathcal{X}|$ itself for a variety of reasons, including computational tractability and generalisation properties of the function approximator. It

is therefore pertinent to ask whether the guarantee established in Corollary 18 can be improved to guarantee embeddabilty in a lower-dimensional Euclidean space. While exact embeddability in lower-dimensional spaces is not always possible, the Johnson–Lindenstrauss lemma (Johnson & Lindenstrauss, 1984) can be used to establish the following result, which shows that lower-dimensional embeddings of $\Pi U^\pi$ are possible, as long as we are prepared to accept a certain level of distortion of the original metric.

**Theorem 19.** *Let $\pi$ be a policy, and $\sim_\pi$ be the equivalence relation on $\mathcal{X}$ defined by $x \sim_\pi y \iff d_{ks}^\pi(x,y) = 0$. For any given $\varepsilon \in (0,1)$, if $m \geq 8\log(|\mathcal{X}/\sim_\pi|)/\varepsilon^2$, where $\mathcal{X}/\sim_\pi$ denotes the quotient set with respect to the equivalence relation $\sim_\pi$, then there exists an embedding $\phi : \mathcal{X} \to \mathbb{R}^m$ such that for all $x,y \in \mathcal{X}$,*

$$(1-\varepsilon)\,d_{ks}^\pi(x,y) \leq \|\phi(x) - \phi(y)\|_2^2 \leq (1+\varepsilon)\,d_{ks}^\pi(x,y),$$

*or equivalently,*

$$(1-\varepsilon)\,\Pi U^\pi(x,y) \leq \|\phi(x) - \phi(y)\|_2^2 \leq (1+\varepsilon)\,\Pi U^\pi(x,y).$$

*Proof.* We recall that the RKHS $\mathcal{H}_{k^\pi}$ is defined by the formula

$$\mathcal{H}_{k^\pi} = \operatorname{span}\{k(x,\cdot) \,:\, x \in \mathcal{X}\},$$

where we do not need to take the completion, as there are only finitely many $x \in \mathcal{X}$, hence $\mathcal{H}_{k^\pi}$ is finite dimensional inner product space, and therefore complete. $\mathcal{H}_{k^\pi}$ is in general a semi inner product space, and to resolve this we take the quotient $\mathcal{H}_{k^\pi}/\sim_\pi$. The set of equivalence classes of $\sim_\pi$ is equivalent to the kernel of the seminorm $\|\cdot\|_{\mathcal{H}_{k^\pi}}$, and in particular is a finite-dimensional vector space, so $\mathcal{H}_{k^\pi}/\sim_\pi$ is a proper inner product space. Let $l = \dim(\mathcal{H}_{k^\pi}/\sim_\pi)$, and let $f_1, f_2, \ldots, f_l$ be an orthonormal basis for $\mathcal{H}_{k^\pi}/\sim_\pi$. We will begin by showing that $(\mathcal{H}_{k^\pi}/\sim_\pi, \|\cdot\|_{\mathcal{H}_{k^\pi}})$ can be isometrically embedded into $(\mathbb{R}^l, \|\cdot\|_2)$. To see this, let $\mathcal{I} : \mathcal{H}_{k^\pi}/\sim_\pi \to \mathbb{R}^l$ be defined by

$$\mathcal{I}(f_j) = e_j \ \forall j \in [l], \ \ \mathcal{I}\left(\sum_{j\in[l]} a_j f_j\right) = \sum_{j\in[l]} a_j\, \mathcal{I}(f_j)$$

We can now see that $\mathcal{I}$ is a Hilbert space isomorphism: for all $i, j \in [l]$,

$$\langle f_i, f_j \rangle_{\mathcal{H}_{k^\pi}} = \langle e_i, e_j \rangle_{\mathbb{R}^l} = \langle\, \mathcal{I}(f_i), \mathcal{I}(f_j)\,\rangle_{\mathbb{R}^l}.$$

Letting $n = |\mathcal{X}/\sim_\pi|$, we can enumerate $\mathcal{X}/\sim_\pi$ as $x_1, \ldots, x_n$, and from the above isometry we know that $\{x_1, \ldots, x_n\} \subset \mathcal{X}$ can be embedded as $\{\mathcal{I}(\varphi^\pi(x_1)), \ldots, \mathcal{I}(\varphi^\pi(x_n))\} \subset \mathbb{R}^l$ with no distortion. Let us use the notation $y_k = \mathcal{I}(\varphi^\pi(x_k))$ as the embedding of $x_k$ into $\mathbb{R}^l$ for brevity.

We can now apply the Johnson-Lindenstrauss lemma: for any $\varepsilon \in (0,1)$, $m \geq \frac{8}{\varepsilon^2}\log(n)$, there exists a linear map $f : \mathbb{R}^l \to \mathbb{R}^m$ such that for any $i, j \in [n]$,

$$(1-\varepsilon)\,\|y_i - y_j\|_{\mathbb{R}^l}^2 \leq \|f(y_i) - f(y_j)\|_{\mathbb{R}^m}^2 \leq (1+\varepsilon)\,\|y_i - y_j\|_{\mathbb{R}^l}^2.$$

The desired statement now follows from rearranging the formula for $n$ and using the isometry

$$\|y_i - y_j\|_{\mathbb{R}^l}^2 = \|\mathcal{I}(\varphi^\pi(x_i)) - \mathcal{I}(\varphi^\pi(x_j))\|_{\mathbb{R}^l}^2 = \|\varphi^\pi(x_i) - \varphi^\pi(x_j)\|_{\mathcal{H}_{k^\pi}}^2 = d_{ks}^\pi(x_i, x_j) = \Pi U^\pi(x_i, x_j),$$

and taking the map $\phi : \mathcal{X}/\sim_\pi \to \mathbb{R}^m$ to be the composition $f \circ \mathcal{I} \circ \varphi^\pi$. $\qquad\square$

**Remark 20.** *The spaces and maps used in the previous proof can be summarized in the following commutative diagram:*

$$\begin{array}{ccc} \mathcal{X}/\!\sim_\pi & \xrightarrow{\ \phi\ } & \mathbb{R}^m \\ {\scriptstyle \varphi^\pi}\Big\downarrow & & \Big\uparrow{\scriptstyle f} \\ \mathcal{H}_{k_\pi} & \xrightarrow{\ \mathcal{I}\ } & \mathbb{R}^l \end{array}$$

**Remark 21.** *Observe that we can improve over the result in Corollary 18 if we accept an error $\epsilon$, so long as $\log(|\mathcal{X}|) < \varepsilon^2 |\mathcal{X}|/8$, in the sense that the result concerns a lower-dimensional embedding.*

### 3.5 Learnable parameterizations

Despite the theoretical appeal of the discussed distances, one of the motivating forces for our work is their applicability to online reinforcement learning with neural networks. Specifically, we are interested in using the derived metrics as a means to learn embeddings $\phi$ that can speed up learning value functions for control. Towards this end, Castro et al. (2021) proposed approximating the diffuse metric $U^\pi$ via $U_\omega$, parameterized as follows:

$$U_\omega(x, y) := \frac{\|\phi_\omega(x)\|_2^2 + \|\phi_\omega(y)\|_2^2}{2} + \beta\theta(\phi_\omega(x), \phi_\omega(y))$$

where $\phi$ is the learned representation parameterized by $\omega$, $\theta(\phi(x), \phi(y))$ is the angular distance between vectors $\phi(x)$ and $\phi(y)$, and $\beta \in (0, \infty)$ is a hyperparameter that weighs the importance of the angular distance.

A peculiar aspect of their method is that this parameterization is approximating the *diffuse metric*, yet when using the representations $\phi_\omega$ for control, they are implicitly making use only of the weighted angular distance, which can be viewed as the *reduced* MICo distance:

$$\beta\theta(\phi_\omega(x), \phi_\omega(y)) \approx \Pi U^\pi(x, y).$$

Although producing strong empirical performance, this results in a somewhat awkward dynamic: the metric space on which the MICo loss is being optimized is different than the metric space where the representations used for control exist. Given the equivalence demonstrated in Section 3.2, we can alleviate this by learning a kernel between representations in the *same inner product space* used for control.

We parametrise the kernel $k^\pi(x, y)$ using the natural inner product on $\mathbb{R}^m$, written as

$$k_\omega(x, y) = \langle \phi_\omega(x), \phi_\omega(y) \rangle \tag{2}$$

From this parametrisation, we can convert the kernel update operator $T_k^\pi$ into a learning target in a similar manner as was done by Castro et al. (2021). Specifically, given two transitions $(x, r_x, x')$ and $(y, r_y, y')$ sampled from the replay buffer, we can define $T_{\bar\omega}^k(r_x, x', r_y, y') = 1 - \frac{1}{2}|r_x - r_y| + \gamma k_{\bar\omega}(x', y')$. Here, $\bar\omega$ is a separate copy of the parameters $\omega$ that are updated less frequently (as suggested by Mnih et al. (2015) and used by Castro et al. (2021)). Our kernel-based loss is then:

$$\mathcal{L}_{\mathrm{KSMe}}(\omega) = \mathbb{E}_{\langle x, r_x, x'\rangle, \langle y, r_y, y'\rangle}\left[\left(T_{\bar\omega}^k(r_x, x', r_y, y') - k_\omega(x, y)\right)^2\right]. \tag{3}$$

Note that this squared semi-gradient loss is taken directly from what was used by MICo (Castro et al., 2021), which was based on the loss originally used by DQN (Mnih et al., 2015). Further details on theoretical properties of this loss in the general context of reinforcement learning can be found in Bertsekas & Tsitsiklis (1996), and discussion in the specific context of metric learning is given by Castro et al. (2021).

With this parametrisation, the KSMe distance between two points $\phi_\omega(x)$ and $\phi_\omega(y)$ is

$$k_\omega(x,x) + k_\omega(y,y) - 2k_\omega(x,y) = \|\phi_\omega(x)\|_2^2 + \|\phi_\omega(y)\|_2^2 - 2\langle \phi_\omega(x), \phi_\omega(y)\rangle$$
$$= \|\phi_\omega(x) - \phi_\omega(y)\|_2^2.$$

That is, the parametrised KSMe distance is exactly the Euclidean distance between the embeddings. This parametrisation is supported by the theory of Section 3.4, as we are approximating the Hilbert space $\mathcal{H}_{k^\pi}$ with the Hilbert space $(\mathbb{R}^m, \|\cdot\|_2)$, which can be summarized as:

$$\|\phi(x)\|_2 = k_\omega(x,x) \approx k^\pi(x,x) = \|\varphi^\pi(x)\|_{\mathcal{H}_{k^\pi}},$$
$$\|\phi(x) - \phi(y)\|_2^2 = d_{ks,\omega}^\pi(x,y) \approx d_{ks}^\pi(x,y) = \|\varphi^\pi(x) - \varphi^\pi(y)\|_{\mathcal{H}_k}^2.$$

## 4    Empirical evaluations

Having demonstrated the theoretical equivalence between $d_{ks}^\pi$ and $\Pi U^\pi$, in this section we perform an empirical investigation to validate their equivalence in practice, and return to our original motivation of using state metrics for representation learning. We do so in small domains where we can compute these distance exactly (Section 4.1), and in more complex domains where we use neural networks to approximate the distances (Section 4.2). All the code for these evaluations are available at https://github.com/google-research/google-research/tree/master/ksme.

### 4.1    Empirical insights into KSMe properties

To investigate the bounds discussed in Section 3.3, we empirically investigate the bounds through the use of Garnet MDPs, as done in Castro et al. (2021), where an empirical analysis was conducted to investigate to what degree $d_{ks}^\pi$ (then referred to as $\Pi U^\pi$) violated the value function upper bound (e.g. how often $d_{ks}^\pi(x,y) - |V^\pi(x) - V^\pi(y)|$ was negative). Given Proposition 17, we can now conduct a more precise study: knowing that the reward variance $\text{Var}_\mathcal{R}(\mathcal{P}^\pi)$ controls the amount by which $d_{ks}^\pi(x,y)$ can be greater than $|V^\pi(x) - V^\pi(y)|$, we plot the bound as $\text{Var}_\mathcal{R}(\mathcal{P}^\pi)$ changes. We plot both the minimum and average signed difference $d(x,y) - |V^\pi(x) - V^\pi(y)|$ for $d = \{d_{ks}^\pi, d_\sim^\pi, U^\pi\}$. As can be seen in Figure 1, although $d_{ks}^\pi$ can violate the upper bound (left plot), on average across the Garnet MDPs  it is more informative of the true value difference than $U^\pi$ and $d_\sim^\pi$ are; this result is consistent with the findings of Castro et al. (2021).

Before proceeding to evaluating $d_{ks}^\pi$ in a more complex domain, it is worth taking stock of the implications of Figure 1. When the upper bound property is violated (see left plot), it could result in dissimilar states being mapped closely together; this could lead to *over-generalization*, as discussed in the introduction above. However, the right plot suggests that in practice this does not happen very often, and is in fact more informative than $d_\sim^\pi$ (for which the upper bound property is not violated). Further, the amount of over-generalization one could suffer is proportional to the reward variance of the environment. When taken together, these results suggest $d_{ks}^\pi$ *can* be an effective way of shaping RL representations during learning.

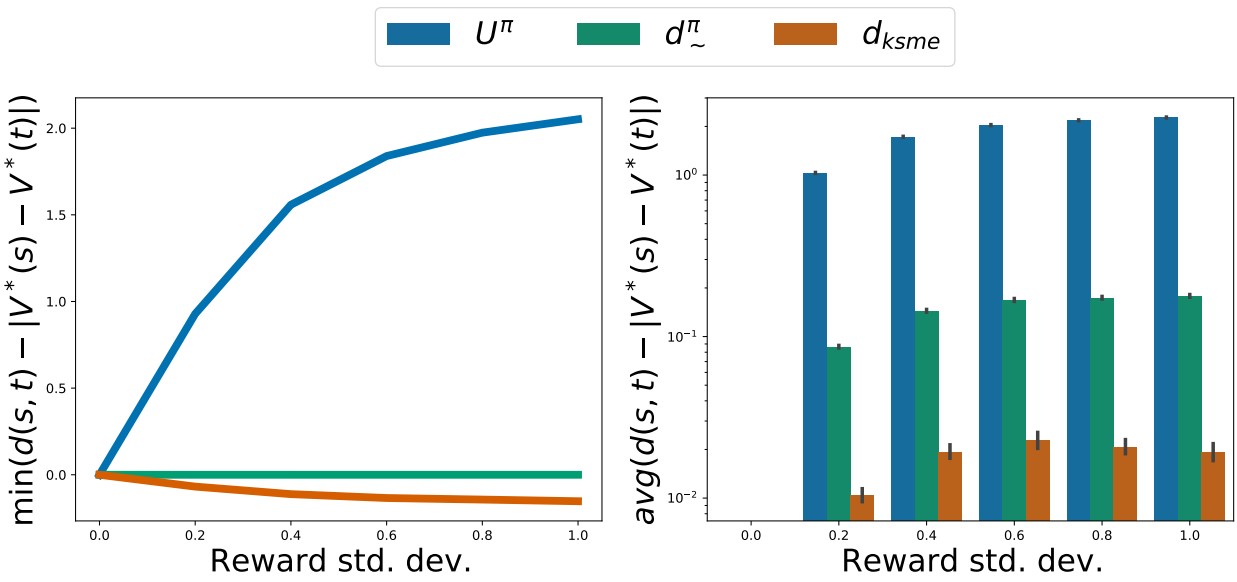

Figure 1: The minimum (left) and average (right) difference between the absolute value function difference $|V^\pi(x) - V^\pi(y)|$ and $U^\pi$ (Castro et al., 2021), $d^\pi_\sim$ (Castro, 2020), and $d^\pi_{ks}$, across 17K random MDPs of varying state and action sizes, as the reward standard deviation $\text{Var}_\mathcal{R}(\mathcal{P}^\pi)$ is increased from 0 to 1.

## 4.2 Large-scale evaluation of KSMe

We adapted the code provided by Castro et al. (2021) to approximate KSMe instead of MICo, incorporating our new loss in the same way. Specifically, we use RL agents that estimate $Q$ values by composing a learned $m$-dimensional representation $\phi_\omega : \mathcal{X} \to \mathbb{R}^m$ (parameterized by $\omega$) with a learned value approximator $\psi_\theta : \mathbb{R}^m \to \mathbb{R}$ (parameterized by $\theta$): $Q(x, \cdot) \approx \psi_\theta(\phi_\omega(x))$.

Given two states $x$ and $y$, the similarity between their representations is expressed via the inner product of their embeddings (as detailed in Equation (2)):

$$k_\omega(x, y) = \langle \phi_\omega(x), \phi_\omega(y) \rangle,$$

We then replace the MICo loss of Castro et al. (2021) ($\mathcal{L}_{\text{MICo}}$) with the $KSMe$ loss:

$$\mathcal{L}_{\text{KSMe}}(\omega) = \mathbb{E}_{\langle x, r_x, x'\rangle, \langle y, r_y, y'\rangle} \left[ \left( T^k_{\bar{\omega}}(r_x, x', r_y, y') - k_\omega(x, y) \right)^2 \right].$$

As done by Castro et al. (2021), the value approximator is trained with the standard temporal difference loss of each agent. In our evaluation, we compared the use of $\mathcal{L}_{\text{KSMe}}$ with $\mathcal{L}_{\text{MICo}}$ on the JAX agents provided by the Dopamine library (Castro et al., 2018): DQN (Mnih et al., 2015), Rainbow (Hessel et al., 2018), QR-DQN (Dabney et al., 2018b), IQN (Dabney et al., 2018a), and M-IQN (Vieillard et al., 2020)[8]. We keep all hyperparameters unchanged from those used by Castro et al. (2021).

Due to the computational expense of running these experiments, we selected four representative Atari 2600 games from the ALE suite (Bellemare et al., 2013) which, we felt, covered the varying

---

[8]Note that these are the same agents on which MICo was originally evaluated.

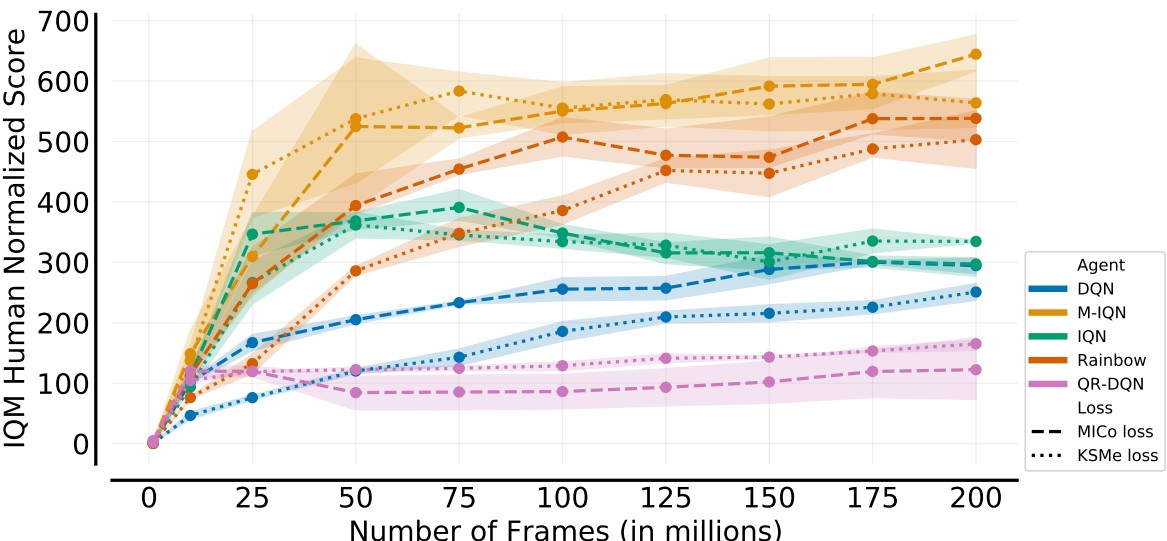

Figure 2: Interquantile mean (Agarwal et al., 2021b) comparison of adding KSMe versus MICo on all the Dopamine (Castro et al., 2018) value-based agents, aggregated over 5 independent runs on four representative games.

dynamics between the original agents and those with the MICo loss. We ran 5 independent seeds for each configuration. In Figure 2 we plot the Interquantile mean (IQM) values which aggregates human-normalized performance across all runs; this metric was introduced by Agarwal et al. (2021b) as a more robust statistic to compare algorithmic performance.

As can be seen, the performance of KSMe is similar to that of MICo, consistent with Theorem 15, which proved their equivalence. Since no hyperparameter optimization was performed for KSMe, it is quite possible the KSMe parameterization can provide empirical gains over MICo. We leave this exploration for future work. In Appendix D we provide the learning curves for each separate agent/game combination.

## 5 Discussion

The empirical results in Section 4 provide experimental validation into the theoretical results of Section 3. Figure 1 follows the trend suggested by Proposition 17, as the worst case difference between the value function difference and $d_{ks}^\pi$ increased approximately linearly with respect to the reward standard deviation. The average gap provides an interesting perspective however, as it demonstrates that on average, as the variance increases, the size of the gap is much larger for $U^\pi$ and $d_\sim^\pi$ than for $d_{ks}^\pi$. Having features which reflect the underlying value function are critical for value-based reinforcement learning, and we hypothesize that this contributes to the empirical success achieved by using KSMe (equivalently, the reduced MICo). As demonstrated in Figure 2, both KSMe and MICo achieve similar empirical performance, which is expected as the underlying distance being learnt is the same. The differences in performance is then a result of learning distances as opposed to kernels in the neural network.

These results also provide further explanation why KSMe (equivalently the reduced MICo) appears to achieve stronger success than bisimulation in deep reinforcement learning settings (the two distances were compared in Castro et al. (2021), where bisimulation was learnt using DBC (Zhang

et al., 2021)). Two possible reasons coming from this work are (i) tighter relationship to the underlying value function and (ii) superior embeddability in neural networks. The hypothesis (i) is supported empirically by Figure 1, and theoretically by the fact that in general $d_{ks}^{\pi}(x, y) \leq d_{\sim}^{\pi}(x, y)$. We believe that this tighter bound leads to improved performance because distances which correctly approximate the value function difference between states allow for straightforward value-based learning. Secondly, the fact that KSMe comes from a Hilbert state structure allows efficient embeddability into low-dimension Euclidean space, a very important concept in deep settings, as this is exactly what neural networks aim to accomplish. Theorem 19 provides a result guaranteeing this is possible. We note that it is not clear whether a similar result for bisimulation metrics is possible, as the Kantorovich metric cannot be approximated in Euclidean spaces with low distortion in general (Peyré & Cuturi, 2019).

## 6 Conclusion

In this work we have taken a kernel perspective on learning representations in reinforcement learning, and introduced a state similarity kernel on Markov decision process state spaces. This kernel naturally induces a distance, which we proved was equal to the reduced MICo distance (Castro et al., 2021). This allowed us to perform a theoretical analysis of the reduced MICo distance which was previously lacking, and answer important questions such as its metric properties and connection to value functions. We then analyzed a previously-unconsidered question: how well the distance itself can be approximated in Euclidean spaces, and prove a bound demonstrating embeddability. We then adapted the loss introduced in Castro et al. (2021) to learn the kernel. While the distance learnt is theoretically equivalent to theirs, our parametrization is theoretically grounded as the neural network embeddings are an approximation to kernel Hilbert space embeddings. To the best of our knowledge, this is the first work which studied *how well* a given state similarity metric can be approximated through a neural network, and provided bounds on the incurred error. These results provide theoretical grounding for the reduced MICo distance, and our kernel perspective analysis may be used in future work to analyze related distances.

It is worth noting that by focusing on the upper-bound properties of behavioural metrics we are providing assymetrical guarantees. Namely, while they can guarantee similar states will have similar values, they do not guarantee that *dissimilar* states will have dissimilar values. It would thus be valuable to investigate further why these metrics can provide such strong performance despite the asymmetrical guarantees.

## Acknowledgements

We thank Gheorghe Comanici for detailed feedback on an earlier version of the paper, and the anonymous TMLR reviewers and action editor Gergely Neu for helping us strengthen our submission.

We would also like to thank the Python community (Van Rossum & Drake Jr, 1995; Oliphant, 2007) for developing tools that enabled this work, including NumPy (Harris et al., 2020), Matplotlib (Hunter, 2007) and JAX (Bradbury et al., 2018).

## Author Contributions

Authors listed in alphabetical order, and individual contributions listed below.

**Pablo Samuel Castro** helped give direction to the work, revised the theoretical results, wrote the code, ran the deep RL experiments, and contributed to writing the paper.

**Tyler Kastner** proved most of the theoretical results, ran the toy experiments, and contributed to writing the paper.

**Prakash Panangaden** helped give direction to the work, revised the theoretical results, supervised Tyler, and contributed to writing the paper.

**Mark Rowland** helped give direction to the work, revised the theoretical results, reviewed the code, and contributed to writing the paper.

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

## A  Extra technical results

For the following two lemmas, we will make use of the sequences $(k_n)_{n\geq0}$, $(U_n)_{n\geq0}$ defined by $k_n \equiv 0$, $k_{n+1} = T_k^\pi(k_n)$, $U_n \equiv 0$, $U_{n+1} = T_M^\pi(U_n)$.

**Lemma 22.** *For any point* $(x_1, x_2, y_1, y_2) \in \mathcal{X}^4$, *and* $n \geq 0$, *we have that*

$$
\mathop{\mathbb{E}}_{\substack{X_1'\sim\mathcal{P}_{x_1}^\pi,X_2'\sim\mathcal{P}_{x_2}^\pi \\ Y_1'\sim\mathcal{P}_{y_1}^\pi,Y_2'\sim\mathcal{P}_{y_2}^\pi}} \left[k_n(X_1', X_2') + k_n(Y_1', Y_2') - 2k_n(X_1', Y_1')\right]
$$

$$
= \mathop{\mathbb{E}}_{\substack{X_1'\sim\mathcal{P}_{x_1}^\pi,X_2'\sim\mathcal{P}_{x_2}^\pi \\ Y_1'\sim\mathcal{P}_{y_1}^\pi,Y_2'\sim\mathcal{P}_{y_2}^\pi}} \left[U_n(X_1', Y_1') - \frac{1}{2}\left(U_n(X_1', X_2') + U_n(Y_1', Y_2')\right)\right].
$$

*Proof.* We show this by induction. Both sides are identically zero at $n = 0$, so we set $n \geq 0$ and assume the induction hypothesis. We can then write out

$$
\mathop{\mathbb{E}}_{\substack{X_1'\sim\mathcal{P}_{x_1}^\pi,X_2'\sim\mathcal{P}_{x_2}^\pi \\ Y_1'\sim\mathcal{P}_{y_1}^\pi,Y_2'\sim\mathcal{P}_{y_2}^\pi}} \left[k_{n+1}(X_1', X_2') + k_{n+1}(Y_1', Y_2') - 2k_{n+1}(X_1', Y_1')\right]
$$

$$
= \mathop{\mathbb{E}}_{\substack{X_1'\sim\mathcal{P}_{x_1}^\pi,X_2'\sim\mathcal{P}_{x_2}^\pi \\ Y_1'\sim\mathcal{P}_{y_1}^\pi,Y_2'\sim\mathcal{P}_{y_2}^\pi}} \left[\left(|r_{X_1}^\pi - r_{Y_1}^\pi| - \frac{1}{2}(|r_{X_1}^\pi - r_{X_2}^\pi| + |r_{Y_1}^\pi - r_{Y_2}^\pi|)\right)\right.
$$

$$
\left. + \mathop{\mathbb{E}}_{\substack{X_1''\sim\mathcal{P}_{X_1'}^\pi,X_2''\sim\mathcal{P}_{X_2'}^\pi \\ Y_1''\sim\mathcal{P}_{Y_1'}^\pi,Y_2''\sim\mathcal{P}_{Y_2'}^\pi}} \left[k_n(X_1'', X_2'') + k_n(Y_1'', Y_2'') - 2k_n(X_1'', Y_1'')\right]\right]
$$

$$
= \mathop{\mathbb{E}}_{\substack{X_1'\sim\mathcal{P}_{x_1}^\pi,X_2'\sim\mathcal{P}_{x_2}^\pi \\ Y_1'\sim\mathcal{P}_{y_1}^\pi,Y_2'\sim\mathcal{P}_{y_2}^\pi}} \left[\left(|r_{X_1}^\pi - r_{Y_1}^\pi| - \frac{1}{2}(|r_{X_1}^\pi - r_{X_2}^\pi| + |r_{Y_1}^\pi - r_{Y_2}^\pi|)\right)\right]
$$

$$
+ \mathop{\mathbb{E}}_{\substack{X_1'\sim\mathcal{P}_{x_1}^\pi,X_2'\sim\mathcal{P}_{x_2}^\pi \\ Y_1'\sim\mathcal{P}_{y_1}^\pi,Y_2'\sim\mathcal{P}_{y_2}^\pi}} \left[\mathop{\mathbb{E}}_{\substack{X_1''\sim\mathcal{P}_{X_1'}^\pi,X_2''\sim\mathcal{P}_{X_2'}^\pi \\ Y_1''\sim\mathcal{P}_{Y_1'}^\pi,Y_2''\sim\mathcal{P}_{Y_2'}^\pi}} \left[k_n(X_1'', X_2'') + k_n(Y_1'', Y_2'') - 2k_n(X_1'', Y_1'')\right]\right]
$$

$$
= \mathop{\mathbb{E}}_{\substack{X_1'\sim\mathcal{P}_{x_1}^\pi,X_2'\sim\mathcal{P}_{x_2}^\pi \\ Y_1'\sim\mathcal{P}_{y_1}^\pi,Y_2'\sim\mathcal{P}_{y_2}^\pi}} \left[\left(|r_{X_1}^\pi - r_{Y_1}^\pi| - \frac{1}{2}(|r_{X_1}^\pi - r_{X_2}^\pi| + |r_{Y_1}^\pi - r_{Y_2}^\pi|)\right)\right]
$$

$$
+ \mathop{\mathbb{E}}_{\substack{X_1'\sim\mathcal{P}_{x_1}^\pi,X_2'\sim\mathcal{P}_{x_2}^\pi \\ Y_1'\sim\mathcal{P}_{y_1}^\pi,Y_2'\sim\mathcal{P}_{y_2}^\pi}} \left[\mathop{\mathbb{E}}_{\substack{X_1''\sim\mathcal{P}_{X_1'}^\pi,X_2''\sim\mathcal{P}_{X_2'}^\pi \\ Y_1''\sim\mathcal{P}_{Y_1'}^\pi,Y_2''\sim\mathcal{P}_{Y_2'}^\pi}} \left[U_n(X_1'', Y_1'') - \frac{1}{2}\left(U_n(Y_1'', Y_2'') + U_n(Y_1'', Y_2'')\right)\right]\right]
$$

$$
= \mathop{\mathbb{E}}_{\substack{X_1'\sim\mathcal{P}_{x_1}^\pi,X_2'\sim\mathcal{P}_{x_2}^\pi \\ Y_1'\sim\mathcal{P}_{y_1}^\pi,Y_2'\sim\mathcal{P}_{y_2}^\pi}} \left[\left(|r_{X_1}^\pi - r_{Y_1}^\pi| - \frac{1}{2}(|r_{X_1}^\pi - r_{X_2}^\pi| + |r_{Y_1}^\pi - r_{Y_2}^\pi|)\right)\right.
$$

$$
\left. + \mathop{\mathbb{E}}_{\substack{X_1''\sim\mathcal{P}_{X_1'}^\pi,X_2''\sim\mathcal{P}_{X_2'}^\pi \\ Y_1''\sim\mathcal{P}_{Y_1'}^\pi,Y_2''\sim\mathcal{P}_{Y_2'}^\pi}} \left[U_n(X_1'', Y_1'') - \frac{1}{2}\left(U_n(Y_1'', Y_2'') + U_n(Y_1'', Y_2'')\right)\right]\right]
$$

$$= \mathop{\mathbb{E}}_{\substack{X_1' \sim \mathcal{P}_{x_1}^\pi, X_2' \sim \mathcal{P}_{x_2}^\pi \\ Y_1' \sim \mathcal{P}_{y_1}^\pi, Y_2' \sim \mathcal{P}_{y_2}^\pi}} \left[ U_{n+1}(X_1', Y_1') - \frac{1}{2}(U_{n+1}(X_1', X_2') + U_{n+1}(Y_1', Y_2')) \right],$$

as desired. $\qquad\square$

**Lemma 14.** *For any measures $\mu$, $\nu$, and $n \geq 0$, we have that*

$$\mathop{\mathbb{E}}_{\substack{X_1, X_2 \sim \mu \\ Y_1, Y_2 \sim \nu}} [k_n(X_1, X_2) + k_n(Y_1, Y_2) - 2k_n(X_1, Y_1)] = \mathop{\mathbb{E}}_{\substack{X_1, X_2 \sim \mu \\ Y_1, Y_2 \sim \nu}} \left[ U_n(X_1, Y_1) - \frac{1}{2}(U_n(X_1, X_2) + U_n(Y_1, Y_2)) \right].$$

*Proof.* We proceed to show this by induction. The base case is straightforward, as both sides are identically zero. We can now assume the induction hypothesis, and can write out

$$\mathop{\mathbb{E}}_{\substack{X_1, X_2 \sim \mu \\ Y_1, Y_2 \sim \nu}} [k_{n+1}(X_1, X_2) + k_{n+1}(Y_1, Y_2) - 2k_{n+1}(X_1, Y_1)]$$

$$= \mathop{\mathbb{E}}_{\substack{X_1, X_2 \sim \mu \\ Y_1, Y_2 \sim \nu}} \left[ \left( |r_{X_1}^\pi - r_{Y_1}^\pi| - \frac{1}{2}(|r_{X_1}^\pi - r_{X_2}^\pi| + |r_{Y_1}^\pi - r_{Y_2}^\pi|) \right) + \gamma \mathop{\mathbb{E}}_{\substack{X_1' \sim \mathcal{P}_{X_1}^\pi \\ X_2' \sim \mathcal{P}_{X_2}^\pi \\ Y_1' \sim \mathcal{P}_{Y_1}^\pi \\ Y_2' \sim \mathcal{P}_{Y_2}^\pi}} [k_n(X_1', X_2') + k_n(Y_1', Y_2') - 2k_n(X_1', Y_1')] \right]$$

$$= \mathop{\mathbb{E}}_{\substack{X_1, X_2 \sim \mu \\ Y_1, Y_2 \sim \nu}} \left[ \left( |r_{X_1}^\pi - r_{Y_1}^\pi| - \frac{1}{2}(|r_{X_1}^\pi - r_{X_2}^\pi| + |r_{Y_1}^\pi - r_{Y_2}^\pi|) \right) + \gamma \mathop{\mathbb{E}}_{\substack{X_1' \sim \mathcal{P}_{X_1}^\pi \\ X_2' \sim \mathcal{P}_{X_2}^\pi \\ Y_1' \sim \mathcal{P}_{Y_1}^\pi \\ Y_2' \sim \mathcal{P}_{Y_2}^\pi}} \left[ U_n(X_1', Y_1') - \frac{1}{2}(U_n(X_1', X_2') + U_n(Y_1', Y_2')) \right] \right] \quad (\star)$$

$$= \mathop{\mathbb{E}}_{\substack{X_1, X_2 \sim \mu \\ Y_1, Y_2 \sim \nu}} \left[ U_{n+1}(X_1, Y_1) - \frac{1}{2}(U_{n+1}(X_1, X_2) + U_{n+1}(Y_1, Y_2)) \right],$$

where $(\star)$ follows from Lemma 22.

$\qquad\square$

## B  Background

### B.1  Markov decision processes

We consider Markov decision processes (MDPs) given by $(\mathcal{X}, \mathcal{A}, \mathcal{P}, \mathcal{R}, \gamma)$, where $\mathcal{X}$ is a finite state space, $\mathcal{A}$ a set of actions, $\mathcal{P} : \mathcal{X} \times \mathcal{A} \to \mathscr{P}(\mathcal{X})$ a transition kernel, and $\mathcal{R} : \mathcal{X} \times \mathcal{A} \to \mathscr{P}(\mathbb{R})$ a reward kernel (where $\mathscr{P}(\mathcal{Z})$ is the set of probability distributions on a measurable set $\mathcal{Z}$). We will write $\mathcal{P}_x^a := \mathcal{P}(x, a)$ for the transition distribution from taking action $a$ in state $x$, $\mathcal{R}_x^a := \mathcal{R}(x, a)$ for the reward distribution from taking action $a$ in state $x$, and write $r_x^a$ for the expectation of this distribution. A policy $\pi$ is a mapping $\mathcal{X} \to \mathscr{P}(\mathcal{A})$. We use the notation $\mathcal{P}_x^\pi = \sum_{a \in \mathcal{A}} \pi(a|x) \mathcal{P}_x^a$ to indicate the state distribution obtained by following one step of a policy $\pi$ while in state $x$. We use $\mathcal{R}_x^\pi = \sum_{a \in \mathcal{A}} \pi(a|x) \mathcal{R}_x^a$ to represent the reward distribution from $x$ under $\pi$, and $r_x^\pi$ to indicate the expected value of this distribution. $\gamma \in [0, 1)$ is the discount factor used to compute the discounted long-term return.

We will often make use of the random trajectory $(X_t, A_t, R_t)_{t \geq 0}$, where $X_t$, $A_t$, and $R_t$ are random variables representing the state, action, and reward at time $t$, respectively. We will occasionally take expectations with respect to policies, written as $\mathbb{E}_\pi[\cdot]$, which should be read as the expectation, given that for all $t \geq 0$ we choose $A_t \sim \pi(\cdot|X_t)$, and receive $R_t \sim \mathcal{R}(\cdot|X_t, A_t)$ and $X_{t+1} \sim \mathcal{P}(\cdot|X_t, A_t)$.

The *value* of a policy $\pi$ is the expected total return an agent attains from following $\pi$, and is described by a function $V^\pi : \mathcal{X} \to \mathbb{R}$, such that for each $x \in \mathcal{X}$,

$$V^\pi(x) = \mathbb{E}_\pi \left[ \sum_{t \geq 0} \gamma^t R_t \,\middle|\, X_0 = x \right],$$

A related quantity is the action-value function $Q^\pi : \mathcal{X} \times \mathcal{A} \to \mathbb{R}$, which indicates the value of taking an action in a state, and then following the policy:

$$Q^\pi(x, a) = \mathbb{E}_\pi \left[ \sum_{t \geq 0} \gamma^t R_t \,\middle|\, X_0 = x, A_0 = a \right].$$

A foundational relationship in reinforcement learning is the *Bellman equation*, which allows the value function of a state to be written recursively in terms of next states. It exists in two forms, for $V^\pi$ and $Q^\pi$ respectively:

$$V^\pi(x) = \mathbb{E}_\pi \left[ R_0 + \gamma V^\pi(X_1) \,\middle|\, X_0 = x \right],$$

$$Q^\pi(x, a) = \mathbb{E}_\pi \left[ R_0 + \gamma V^\pi(X_1) \,\middle|\, X_0 = x, A_0 = a \right].$$

Rewriting these equations without the use of the random trajectory, we have

$$V^\pi(x) = r_x^\pi + \gamma \mathop{\mathbb{E}}_{X' \sim \mathcal{P}_x^\pi} \left[ V^\pi(X') \right],$$

$$Q^\pi(x, a) = r_x^a + \gamma \mathop{\mathbb{E}}_{X' \sim \mathcal{P}_x^a, A' \sim \pi(\cdot|X')} \left[ Q^\pi(X', A') \right].$$

The Bellman operator $T^\pi$ transforms the above equations into an operator over $\mathbb{R}^\mathcal{X}$ (or $\mathbb{R}^{\mathcal{X} \times \mathcal{A}}$ — we will overload the use of $T^\pi$ and let the type signature indicate which is being used), given by

$$T^\pi V(x) = r_x^\pi + \gamma \mathop{\mathbb{E}}_{X' \sim \mathcal{P}_x^\pi} \left[ V(X') \right],$$

$$T^\pi Q(x, a) = r_x^a + \gamma \mathop{\mathbb{E}}_{X' \sim \mathcal{P}_x^a, A' \sim \pi(\cdot|X')} \left[ Q(X', A') \right].$$

Written in this way, we see that $Q^\pi$ and $V^\pi$ are fixed points of $T^\pi$, and with some work one can also see that $T^\pi$ is a contraction with modulus $\gamma$. As a corollary of Banach's fixed point theorem, one can choose $V_0$ arbitrarily and update $V_{k+1} = T^\pi V_k$, and converge to $V^\pi$, this is the algorithm known as *value iteration*.

An optimal policy $\pi^*$ is a policy which achieves the maximum value function at each state, which we will denote $V^*$. It satisfies the Bellman optimality recurrence:

$$V^*(x) = \max_{a \in \mathcal{A}} \left[ R_0 + \gamma V^*(X_1) | X_0 = x \right].$$

## C   Behavioural metrics

In this section, we review various distances which have appeared in literature, and discuss how they relate to each other. A behavioural metric is usually defined using the following pattern. One is comparing the difference between two states, the first and most obvious difference between the states is a reward difference, accordingly this is the first term in the behavioural metric definition. But one wants to take into account the differences in the subsequent evolution of the system starting from the two states. Thus, one needs a notion of difference in the "next states". However, since these are probabilistic systems, there is no unique next state; one has a probability distribution over the next states. Thus, one needs a metric that can measure the differences between probability distributions.

Metrics between probability distrubutions have a rich theory (Rachev et al., 2013) and history. We review parts of this theory in the first subsection before using these metrics to construct behavioural metrics between the states of an MDP.

### C.1   Metrics on probability distributions

#### C.1.1   The Kantorovich metric

Given two probability measures $\mu$ and $\nu$ on a set $\mathcal{X}$, a coupling $\lambda \in \mathscr{P}(\mathcal{X} \times \mathcal{X})$ of the measures is a joint distribution with marginals $\mu$ and $\nu$. Formally, we have that for every measurable subset $A \subset \mathcal{X}$,

$$\lambda(A \times \mathcal{X}) = \mu(A) \text{ and } \lambda(\mathcal{X} \times A) = \nu(A).$$

We define $\Lambda(\mu, \nu)$ to represent the set of all couplings of $\mu$ and $\nu$. This set is non-empty in general, in particular the independent coupling $\lambda = \mu \times \nu$ always exists. Couplings are essential for the definition of the Kantorovich metric $\mathcal{W}$ (Kantorovich & Rubinshtein, 1958) (also known as the Wasserstein metric), a metric on the space of probability distributions on $\mathcal{X}$. Given a metric $d$ on $\mathcal{X}$[9], the Kantorovich metric is defined as

$$\mathcal{W}(d)(\mu, \nu) = \inf_{\lambda \in \Lambda(\mu, \nu)} \int d(x, y) \, \mathsf{d}\lambda(x, y).$$

The coupling which attains the infimum always exists, and is referred to as the *optimal coupling* of $\mu$ and $\nu$ (Villani, 2008).

#### C.1.2   The Łukaszyk–Karmowski distance

We recall that the Kantorovich metric optimizes over the space of all couplings; indeed, the computational difficulty of calculating the Kantorovich distance comes from calculating this infimum, since for each pair of measures one must solve an optimization problem. The *Łukaszyk–Karmowski distance* $d_{\text{ŁK}}$ (Łukaszyk, 2004)avoids this optimization, and instead considers the independent coupling between the measures. That is,

$$d_{\text{ŁK}}(d)(\mu, \nu) = \int d(x, y) \, \mathrm{d}(\mu \times \nu)(x, y),$$

or equivalently

$$d_{\text{ŁK}}(d)(\mu, \nu) = \mathbb{E}_{X \sim \mu, Y \sim \nu}[d(X, Y)].$$

---

[9]To be precise, we require $(\mathcal{X}, d)$ to be a Polish space, but we drop these technical assumptions for clarity of presentation.

Without the need for the optimization over couplings, the computation of $d_{\text{ŁK}}$ reduces to the above computation, which is much more computationally efficient. When the base distance $d$ is the Euclidean distance $\| \cdot \|$, the Łukaszyk–Karmowski distance has been used in econometrics, usually referred to as Gini's coefficient (Gini, 1912; Yitzhaki, 2003).

The computational advantage comes at a price, however. While the Kantorovich metric satisfies all the axioms of a proper metric, the Łukaszyk–Karmowski distance does not. This is due to the fact that measures can have non-negative self distances, meaning one may find a measure $\mu$ such that $d_{\text{ŁK}}(d)(\mu, \mu) > 0$. One can show that a measure $\mu$ satisfies $d_{\text{ŁK}}(d)(\mu, \mu) = 0$ if and only if $\mu$ is a Dirac measure, that is $\mu = \delta_x$ for some $x \in \mathcal{X}$ (Łukaszyk, 2004). Intuitively, $d_{\text{ŁK}}(d)(\mu, \mu)$ should be seen as a measure of *dispersion* of $\mu$. One interpretation of this in the literature (Łukaszyk, 2004) is that $d_{\text{ŁK}}$ captures a concept of *uncertainty*: given two random variables $X \sim \mu$, $Y \sim \nu$, unless $\mu$ and $\nu$ are point masses, observed values of $X$ and $Y$ are less likely to be equal depending on the dispersions of $\mu$ and $\nu$. Hence $d_{\text{ŁK}}$ captures a measure of uncertainty in the observed distance of $X$ and $Y$, compared to a proper probability metric which would assign distance 0 if $X \stackrel{L}{=} Y$.

The concept of distance functions with non-zero self distances has been considered before, in particular through *partial metrics* (Matthews, 1994). A partial metric is a function $d : \mathcal{X} \times \mathcal{X} \to [0, \infty)$ such that for any $x, y, z \in \mathcal{X}$:

- $0 \le d(x, y)$ *Non-negativity*

- $d(x, x) \le d(x, y)$ *Small self-distances*

- $d(x, y) = d(y, x)$ *Symmetry*

- if $d(x, x) = d(x, y) = d(y, y)$, then $x = y$ *Indistancy implies equality*

- $d(x, y) \le d(x, z) + d(y, z) - d(z, z)$ *Modified triangle inequality*

We note that there were additional axioms added to this definition, rather than simply removing the requirement that $d(x, x) = 0$. This is due to the fact that this definition was constructed so that one can easily construct a proper metric $\tilde{d}$ from a partial metric $d$, given by $\tilde{d}(x, y) = d(x, y) - \frac{1}{2}(d(x, x) + d(y, y))$. We can now show that this definition is indeed too strong for the Łukaszyk–Karmowski distance, which we demonstrate in the following examples.

**Example 1.** *The Łukaszyk–Karmowski distance does not have small self-distances.*

*Proof.* Take $\mathcal{X} = [0, 1]$, $d = |\cdot|$, $\mu = \delta_{1/2}$, $\nu = U([0, 1])$. Then one can calculate $d_{\text{ŁK}}(d)(\nu, \nu) = \int_0^1 \int_0^1 |x - y| dx dy = \frac{1}{3}$, and $d_{\text{ŁK}}(d)(\mu, \nu) = \int_0^1 |x - \frac{1}{2}| dx = \frac{1}{4}$. But then we have $d_{\text{ŁK}}(d)(\nu, \nu) = \frac{1}{3} > \frac{1}{4} = d_{\text{ŁK}}(d)(\mu, \nu)$. $\square$

**Example 2.** *The Łukaszyk–Karmowski distance does not satisfy the modified triangle inequality.*

*Proof.* Take $\mathcal{X} = [0, 1]$, $d = |\cdot|$, $\mu = \delta_0$, $\nu = \delta_1$, $\eta = \frac{1}{2}(\delta_0 + \delta_1)$. We can then calculate $d_{\text{ŁK}}(d)(\mu, \nu) = |1 - 0| = 1$, $d_{\text{ŁK}}(d)(\mu, \eta) = d_{\text{ŁK}}(d)(\nu, \eta) = \frac{1}{2}(0) + \frac{1}{2}(1) = \frac{1}{2}$, $d_{\text{ŁK}}(d)(\eta, \eta) = \frac{1}{4}(0) + \frac{1}{2}(1) + \frac{1}{4}(0) = \frac{1}{2}$. Combining, we have $d_{\text{ŁK}}(d)(\mu, \eta) + d(\nu, \eta) - d(\eta, \eta) = \frac{1}{2} + \frac{1}{2} - \frac{1}{2}$, which then gives us

$$d_{\text{ŁK}}(d)(\mu, \nu) = 1 > \frac{1}{2} = d_{\text{ŁK}}(d)(\mu, \eta) + d(\nu, \eta) - d(\eta, \eta),$$

breaking the modified triangle inequality. $\square$

To account for this, Castro et al. (2021) introduced a new notion of distance known as *diffuse metrics*. A diffuse metric is a function $d : \mathcal{X} \times \mathcal{X} \to [0, \infty)$ such that for any $x, y, z \in \mathcal{X}$:

- $0 \leq d(x, y)$                                                *Non-negativity*

- $d(x, y) = d(y, x)$                                         *Symmetry*

- $d(x, y) \leq d(x, z) + d(y, z)$                           *Triangle inequality*

It is straightforward to see that the Łukaszyk–Karmowski distance is a diffuse metric. In addition, an attractive property of the Łukaszyk–Karmowski distance for the reinforcement learning setting is that it lends itself readily to stochastic approximation. Given two independent streams of samples $(x_n)_{n \geq 1}$, $(y_n)_{n \geq 1}$ from random variables $X \sim \mu$ and $Y \sim \nu$, a base metric $d$ such that $d(X, Y)$ has finite variance, and a sequence of step sizes $(\alpha_n)_{n \geq 1}$ satisfying the Robbins-Monro conditions, one can construct a sequence of iterates $(d_n)$ defined by

$$d_n = (1 - \alpha_n) \, d_{n-1} + \alpha_n \, d(x_n, y_n),$$

with $d_0 = 0$. Then we have $d_n \to d_{\text{ŁK}}(d)(\mu, \nu)$ as $n \to \infty$ (Robbins & Monro, 1951).

## C.2 Bisimulation metrics

### C.2.1 Bisimulation relations

Bisimulation was invented in the context of concurrency theory by Milner (1980) and Park (1981). Probabilistic bisimulation (Larsen & Skou, 1991; Blute et al., 1997; Desharnais et al., 2002; Panangaden, 2009) (henceforth just called bisimulation) is an equivalence on the state space of a labelled Markov process, where two states are considered equivalent if the behaviour from the states are *indistinguishable*. To define indistinguishability, we demand that transition probabilities to equivalence classes should be the same for equivalent states; that is, the equivalence classes preserve the dynamics of the process. In addition if there are more observables, for example, rewards, those should match as well. Bisimulation for MDPs was defined in Givan et al. (2003). This intuition can now be transformed into a definition.

**Definition 23.** *An equivalence relation $R$ on $\mathcal{X}$ is a bisimulation relation if*

$$xRy \implies \forall a \in A, r_x^a = r_y^a \text{ and } \forall C \in \mathcal{X}/R, \mathcal{P}_x^a(C) = \mathcal{P}_y^a(C).$$

We say that states $x$ and $y$ are bisimilar if there exists a bisimulation relation $R$ such that $xRy$. We remark that there exists at least one bisimulation relation, as the diagonal relation $\Delta = \{(x, x) : x \in \mathcal{X}\}$ is always a bisimulation relation, albeit the least interesting one. One refers to the largest bisimulation relation as $\sim$, which is often the one of interest. This version is due to Larsen and Skou (Larsen & Skou, 1991) and the extension to continuous state spaces is due to (Blute et al., 1997; Desharnais et al., 2002).

As Markov decision processes may be seen as Markov processes with rewards, bisimulation relations and metrics have a natural analogue in this setting.

### C.2.2 Bisimulation metrics

The stringency of bisimulation relations is apparent in the MDP case: if two states have bisimilar transition dynamics, that is we have that $\forall a$ and $\forall C \in \mathcal{X}/R$, $\mathcal{P}_x^a(C) = \mathcal{P}_y^a(C)$, but $|r_x^a - r_y^a| = \varepsilon > 0$,

then $x$ and $y$ are in different equivalence classes. This motivates us to introduce a metric analogue of bisimulation. We will write $\mathcal{M}(\mathcal{X})$ to represent the space of bounded pseudometrics on $\mathcal{X}$.

**Theorem 24** (Ferns et al. (2004)). *Define $\mathcal{F} : \mathcal{M}(\mathcal{X}) \to \mathcal{M}(\mathcal{X})$ as*

$$\mathcal{F}(d)(x, y) = \max_{a \in \mathcal{A}} \left( |r_x^a - r_y^a| + \gamma \, \mathcal{W}(d)(\mathcal{P}_x^a, \mathcal{P}_y^a) \right).$$

*Then $\mathcal{F}$ is a contraction in $\|\cdot\|_\infty$ with modulus $\gamma$, and hence exhibits a unique fixed point $d_\sim$, which we denote the bisimulation metric.*

Justification for the term bisimulation metric follows from the fact that the kernel (the kernel of a pseudometric $d$ is the set of pairs of points deemed 'equivalent', formally the set $\{x, y : d(x, y) = 0\}$) of $d_\sim$ is a bisimulation relation.

**Proposition 25** (Ferns et al. (2004)). *$V^*$ is $1$-Lipschitz with respect to $d_\sim$, that is for any $x, y \in \mathcal{X}$,*

$$|V^*(x) - V^*(y)| \leq d_\sim(x, y).$$

### C.3  $\pi$-**bisimulation metrics**

Bisimulation considers equivalence across all possible actions, which is a strong notion of equivalence. In many settings, in a given state an agent may not be concerned with the behaviour under every possible action, but instead only with the actions which it may take under a given policy. On-policy bisimulation (Castro, 2020) was introduced to address this. The definition is a straightforward modification of bisimulation, adapted to a given policy.

**Definition 26** (On-policy bisimulation relations). *Let $\pi$ be a fixed policy. An equivalence relation $R$ on $\mathcal{X}$ is a $\pi$-bisimulation relation if*

$$xRy \implies r_x^\pi = r_y^\pi \text{ and } \forall C \in \mathcal{X}/R, \mathcal{P}_x^\pi(C) = \mathcal{P}_y^\pi(C).$$

**Remark 27.** *It is important to note that while the definitions of bisimulation relations and $\pi$-bisimulation relations appear very similar, they have intrinsic differences. In particular, two states which are bisimilar need not be $\pi$-bisimilar, and two states which are $\pi$-bisimilar need not be bisimilar. To see this intuitively, consider two states which are bisimilar, then any action from either state obtains the same reward and transitions to bisimulation equivalence classes with equal probabilities. But a given policy may select actions differently between the two states so that the expected rewards under the policy are different between the states, and in particular the two states are not $\pi$-bisimilar. On the other hand, two states may have different dynamics across different actions, and hence not be bisimilar, but the policy can balance the actions such that the states are $\pi$-bisimilar.*

The $\pi$-bisimilarity equivalence relation, like ordinary bisimulation, is sensitive to small changes in the system parameters; so defining a metric in place of a relation is the natural next step.

**Theorem 28** (Castro (2020)). *For a policy $\pi$, define $\mathcal{F}^\pi : \mathcal{M}(\mathcal{X}) \to \mathcal{M}(\mathcal{X})$ as*

$$\mathcal{F}^\pi(d)(x, y) = |r_x^\pi - r_y^\pi| + \gamma \mathcal{W}(d)(\mathcal{P}_x^\pi, \mathcal{P}_y^\pi).$$

*Then $\mathcal{F}^\pi$ is a contraction in $\|\cdot\|_\infty$ with modulus $\gamma$, and hence admits a unique fixed point $d_\sim^\pi$, which we define to be the $\pi$-bisimulation metric.*

It is straightforward to see that the kernel of $d_\sim^\pi$ is an on-policy bisimulation relation, justifying its name. Akin to bisimulation metrics, $\pi$-bisimulation metrics possess desirable continuity properties when it comes to *policy* value functions.

**Proposition 29** ([Castro](#) ([2020](#))). *Let $\pi$ be any policy, then $V^\pi$ is 1-Lipschitz with respect to $d_\sim^\pi$, that is for any $x, y \in \mathcal{X}$,*

$$|V^\pi(x) - V^\pi(y)| \leq d_\sim^\pi(x, y).$$

### C.3.1 Learning bisimulation metrics

While bisimulation metrics come from a rich theoretical background, they lack application in practice due to the difficulty of learning them in online settings, which is desirable for many representation learning purposes. If $\mathcal{P}$ and $\mathcal{R}$ were known exactly, then $\mathcal{F}^\pi$ can be repeatedly applied in a dynamic programming fashion, and will converge as it is a contractive map. However, when $\mathcal{P}$ and $\mathcal{R}$ are unknown and only samples are available, learning $d_\pi^\sim$ becomes troublesome, as estimates for $\mathcal{W}$ are generally biased and result in learning different fixed points ([Ferns et al., 2006](#); [Comanici et al., 2012](#)).

### C.4 Deep bisimulation for control

[Zhang et al.](#) ([2021](#)) propose a method of learning $\pi$-bisimulation metrics in representation space. They train a Gaussian dynamics model $\hat{\mathcal{P}}$ to approximate $\mathcal{P}$, and learn an encoder $\phi : \mathcal{X} \to \mathbb{R}^d$. They train $\phi$ so that representation distances approximate $\pi$-bisimulation distances, and use gradient descent to train

$$\|\phi(x) - \phi(y)\|_1 \approx |r_x^\pi - r_y^\pi| + \gamma \mathcal{W}_2(\|\cdot\|_2)(\widehat{\mathcal{P}_x^\pi}, \widehat{\mathcal{P}_y^\pi}).$$

The choice of $\mathcal{W}_2(\|\cdot\|_2)$ and Gaussian transitions $\hat{\mathcal{P}}$ is for computational efficiency, as

$$\mathcal{W}_2(\|\cdot\|_2)(\mathcal{N}(\mu_1, \Sigma_1), \mathcal{N}(\mu_2, \Sigma_2)) = \|\mu_1 - \mu_2\|_2^2 + \left\|\Sigma_1^{1/2} - \Sigma_2^{1/2}\right\|_F^2,$$

where $\|\cdot\|_F$ is the Frobenius norm. As noted by [Kemertas & Aumentado-Armstrong](#) ([2021](#)), this computational advantage widened the theory-practice gap, as (i) it was not proven whether a $\pi$-bisimulation existed when $\mathcal{W}_2$ was used (this was since proven in [Kemertas & Jepson](#) ([2022](#))), (ii) the $L_1$ norm was used for representation distances but the $L_2$ norm was used for the base metric of $\mathcal{W}_2$, and (iii) a dynamics model was used instead of the ground truth dynamics.

### C.5 Background on reproducing kernel Hilbert spaces

We begin by reviewing mathematical background covering vector spaces, reproducing kernel Hilbert spaces, the MMD, and its equivalence to the energy distance.

### C.5.1 Hilbert spaces

A (real) normed space is a vector space $V$ with a function $\|\cdot\| : V \to \mathbb{R}$, which satisfies the following for all $x, y \in V$, $\alpha \in \mathbb{R}$:

- $\|x\| \geq 0$ *Positivity*

- $\|x\| = 0 \iff x = 0$ *Identity of indiscernibles*

- $\|\alpha x\| = |\alpha|\|x\|$ *Absolute homogeneity with respect to scalar multiplication*

- $\|x + y\| \leq \|x\| + \|y\|$ *Triangle inequality*

A normed space is a stronger notion than a metric space, since a norm induces a metric $d$ through $d(x, y) = \|x - y\|$. An inner product space is a stronger notion of a normed space, which is described as a vector space $V$ with a function $\langle \cdot, \cdot \rangle : V \times V \to \mathbb{R}$ such that for all $x, y, z \in V$, $\alpha, \beta \in \mathbb{R}$:

- $\langle x, y \rangle = \langle y, x \rangle$                    *Symmetry*

- $\langle x, \alpha y + \beta z \rangle = \alpha \langle x, y \rangle + \beta \langle x, z \rangle$        *Linearity in the first argument*

- $\langle x, x \rangle \geq 0$ and $\langle x, x \rangle = 0 \iff x = 0$      *Positive definiteness*

An inner product induces a normed space through $\|x\| = \langle x, x \rangle^{1/2}$. If an inner product space induces a normed space whose topology is complete, then the space $(V, \langle \cdot, \cdot \rangle)$ is referred to as a *Hilbert space*. A normed space whose topology is complete is referred to as a *Banach space*, and we remark that Hilbert spaces are a proper subset of Banach spaces.

Hilbert spaces have many desirable properties, and one which will become important for the following theory is the *Riesz representation theorem* (Riesz, 1907). Given a Hilbert space $V$, a map $T : V \to \mathbb{R}$ is linear if:

$$T(\alpha x + \beta y) = \alpha\, T(x) + \beta\, T(y) \text{ for all } x, y \in V, \alpha, \beta \in \mathbb{R}.$$

Continuity is easy to verify for linear maps: a linear map $T$ is continuous if and only if $T$ is bounded, meaning that there exists $C \in \mathbb{R}$ such that

$$\|T(x)\| \leq C\|x\|, \text{ for all } x \in V.$$

The set of all continuous linear operators on $V$ is known as the dual space of $V$, and often referred to as $V^\star$. The Riesz representation theorem states that if $V$ is a Hilbert space, then $V$ and $V^\star$ are isometrically isomorphic. Equivalently, this means that for any linear operator $T : V \to \mathbb{R}$, there exists a unique $x_T \in V$ such that

$$\langle x, x_T \rangle = T(x) \text{ for all } x \in V.$$

The interested reader can refer to any text on functional analysis for further details, such as Rudin (1974).

### C.5.2 Reproducing kernel Hilbert spaces and the MMD

Let $\mathcal{X}$ be a finite set and $\mathcal{H}$ be a Hilbert space of real functions on $\mathcal{X}$. For a point $x \in \mathcal{X}$, the evaluation functional $L_x : \mathcal{H} \to \mathbb{R}$ is defined by

$$L_x(f) = f(x).$$

If $L_x$ is a continuous functional for all $x \in \mathcal{X}$, we say that $\mathcal{H}$ is a *reproducing kernel Hilbert space* (RKHS) (Schölkopf et al., 2018; Aronszajn, 1950). Suppose $\mathcal{H}$ is a reproducing kernel Hilbert space, then for each $x \in \mathcal{X}$, $L_x$ is linear and continuous, and the Riesz representation theorem implies that there exists a unique $k_x \in \mathcal{H}$ such that

$$L_x(f) = \langle f, k_x \rangle_{\mathcal{H}}.$$

Since $k_x \in \mathcal{H}$, we can write

$$k_x(y) = L_y(k_x) = \langle k_x, k_y \rangle_{\mathcal{H}}.$$

This is used to define the *reproducing kernel $k$* of $\mathcal{H}$ as $k(x,y) = \langle k_x, k_y \rangle_{\mathcal{H}}$. We will sometimes write $k_{\mathcal{H}}$ to emphasize the dependence of the kernel on the Hilbert space. One can note that the functions $k_x$ and $k_y$ above can be recovered as the kernel fixed at a single point, that is $k_x = k(x, \cdot) \in \mathcal{H}$, and $k_y = k(y, \cdot) \in \mathcal{H}$. This is where the *reproducing property* comes from, as we see that $k$ 'reproduces' itself:

$$k(x,y) = \langle k(x, \cdot), k(y, \cdot) \rangle_{\mathcal{H}}.$$

In the previous paragraphs, we began with a Hilbert space of functions whose evaluation functional was continuous and obtained a reproducing kernel for this space. On the other hand, in Section 2.3 we took the opposite direction: we began with a positive definite kernel on a set and constructed a Hilbert space of functions, the equivalence of these two approaches is known as the *Moore-Aronszajn theorem* (Aronszajn, 1950).

Let us recall the definition of the MMD introduced earlier in the main text:

**Definition 6** (Gretton et al. (2012))**.** *Let $k$ be a kernel on $\mathcal{X}$, and $\Phi : \mathscr{P}(\mathcal{X}) \to \mathcal{H}_k$ be as defined above. Then the Maximum Mean Discrepancy (MMD) is a pseudometric on $\mathscr{P}(\mathcal{X})$ defined by*

$$MMD(k)(\mu, \nu) = \|\Phi(\mu) - \Phi(\nu)\|_{\mathcal{H}_k}.$$

The MMD can also be seen as arising from a lifting of kernels on $\mathcal{X}$ onto kernels on $\mathscr{P}(\mathcal{X})$ (Guilbart, 1979). Given a kernel $k$ on $\mathcal{X}$, define $K(\mu, \nu)$ for $\mu, \nu \in \mathscr{P}(\mathcal{X})$ as

$$K(\mu, \nu) = \langle \Phi(\mu), \Phi(\nu) \rangle_{\mathcal{H}_k} = \int_{\mathcal{X} \times \mathcal{X}} k(x,y) \, d(\mu \otimes \nu)(x,y).$$

It is immediate that $K$ retains all properties of being a positive definite kernel as it arises from the inner product $\langle \cdot, \cdot \rangle_{\mathcal{H}_k}$. The MMD can then be seen as the metric $\rho_k$ on $\mathscr{P}(\mathcal{X})$. We remark that the MMD with $K$ allows one to metrize $\mathscr{P}(\mathscr{P}(\mathcal{X}))$, but we do not need this in this work.

One can also show that the MMD is an integral probability metric (Müller, 1997), since we can show that

$$\mathrm{MMD}(k)(\mu, \nu) = \sup_{f \in \mathcal{H}_k : \|f\|_{\mathcal{H}_k} \le 1} \left| \int_{\mathcal{X}} f \, d\mu - \int_{\mathcal{X}} f \, d\nu \right|.$$

To see that this corresponds to the MMD as defined above, one can write out

$$
\begin{aligned}
\sup_{f \in \mathcal{H}_k : \|f\|_{\mathcal{H}_k} \le 1} \left| \int_{\mathcal{X}} f \, d\mu - \int_{\mathcal{X}} f \, d\nu \right| &= \sup_{f \in \mathcal{H}_k : \|f\|_{\mathcal{H}_k} \le 1} |\langle f, \Phi(\mu) \rangle_{\mathcal{H}_k} - \langle f, \Phi(\nu) \rangle_{\mathcal{H}_k}| \\
&= \sup_{f \in \mathcal{H}_k : \|f\|_{\mathcal{H}_k} \le 1} |\langle f, \Phi(\mu) - \Phi(\nu) \rangle_{\mathcal{H}_k}| \\
&= \|\Phi(\mu) - \Phi(\nu)\|_{\mathcal{H}_k},
\end{aligned}
$$

where we used the following fact for general Hilbert spaces $\mathcal{H}$: $\sup_{x : \|x\|_{\mathcal{H}} \le 1} \langle x, y \rangle_{\mathcal{H}} = \|y\|_{\mathcal{H}}$, which follows from the Cauchy-Schwarz inequality.

# D    Extra empirical results

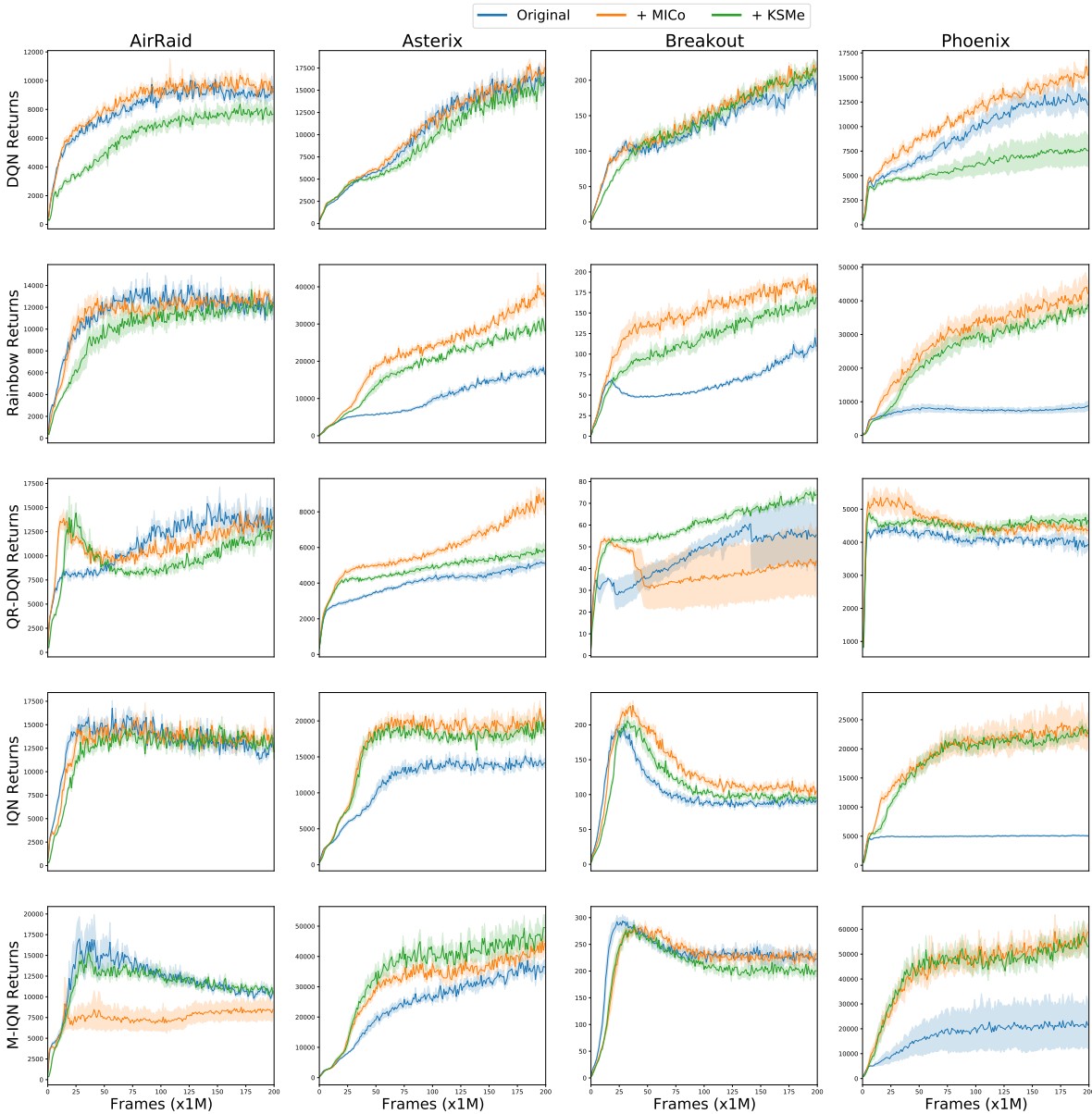

Figure 3: Comparison of adding KSMe versus MICo on all the Dopamine (Castro et al., 2018) value-based agents, on four representative games. Solid lines represent the average over 5 independent runs, while the shaded areas represent 75% confidence intervals.

