# OpenReview forum: "A Kernel Perspective on Behavioural Metrics for Markov Decision Processes"
_TMLR — Accepted by TMLR_

### Review · Reviewer_ea48 · 2023-01-04

**Summary Of Contributions:**

The paper studies the problem of defining behavioral metrics for a given MDP. A behavioral metric for a given policy is a quantity that for any two states attempts to give an upper bound on the difference of their values under the same policy. In this paper, the authors provide a new perspective on this problem using the RKHS theory. They show that one can use the reward function and the transition kernel of the MDP under the given policy to define a kernel that gives rise to a metric. Given this kernel, the authors argue that the feature embedding of the states in RKHS is useful for value function learning.

The authors build their work upon the prior work of Castro et al. (2021) which does not discuss the RKHS view but rather only the metric view of the behavioral metric. The current work fills the gaps of the prior work and provides a behavioral metric that upper-bounds the difference of the value functions.

Borrowing the idea from the fixed policy setting (the theory developed in this paper), the authors propose to take their feature embedding method to the online RL setting where they use neural networks to learn the feature embedding, resulting in performance improvement over base methods such as DQN.

**Audience:**

Yes

**Broader Impact Concerns:**

There is no concern about the ethical implication of significant matters in this work.

**Claims And Evidence:**

Yes

**Requested Changes:**

Please motivate in more depth the importance of the behavioral metric.

**Strengths And Weaknesses:**

Strengths:
1- the paper is well written
2- The theory is solid
3- The paper supports the presented claims with both theory and empirical study

Weakness:
1- The paper relies too much on the narrative of the prior work of Castro et al. (2021) on the importance of the notion of behavioral metrics.
The authors are encouraged to provide more arguments on why such a thing might matter. Reading the paper at its current stage, I am still not fully following why such a thing is important.

---

### Review · Reviewer_bNqr · 2023-01-11

**Summary Of Contributions:**

The authors consider the problem of defining and evaluating similarity measures between states in MDPs. Some theoretical tools are developed, and numerical approximations, whose practical performance is studied.

**Audience:**

Yes

**Broader Impact Concerns:**

None.

**Claims And Evidence:**

Yes

**Requested Changes:**

See above.

**Strengths And Weaknesses:**

The authors appear unaware of a large body of literature related to the application of reproducing kernel Hilbert space theory to Markov Decision Problems. In particular, see:

Ormoneit, D., & Sen, Ś. (2002). Kernel-based reinforcement learning. Machine learning, 49(2), 161-178.

Lever, G., Shawe-Taylor, J., Stafford, R., & Szepesvári, C. (2016, February). Compressed conditional mean embeddings for model-based reinforcement learning. In Proceedings of the AAAI Conference on Artificial Intelligence (Vol. 30, No. 1).

Farahmand, A. M., Ghavamzadeh, M., Szepesvári, C., & Mannor, S. (2016). Regularized policy iteration with nonparametric function spaces. The Journal of Machine Learning Research, 17(1), 4809-4874.

Yang, Z., Jin, C., Wang, Z., Wang, M., & Jordan, M. (2020). Provably efficient reinforcement learning with kernel and neural function approximations. Advances in Neural Information Processing Systems, 33, 13903-13916.

Koppel, A., Warnell, G., Stump, E., Stone, P., & Ribeiro, A. (2020). Policy evaluation in continuous MDPs with efficient kernelized gradient temporal difference. IEEE Transactions on Automatic Control, 66(4), 1856-1863.

In particular, in the aforementioned references, the proposed similarity measure between states have has been previously considered in a simplified form in prior works, with the explicit goal of approximating a value (or Q) function that belongs to an RKHS. Therefore, the first bullet on page 2 is a dubious claim, as this notion of similarity pre-dates this work. The authors have not done a thorough literature review on the use of kernel methods in RL. Moreover, the reason for adding d_2 in the definition of the distance is not very clear, and it additionally cannot be easily evaluated, making it not very useful for algorithmic practical use.

How does the proposed similarity measure actually improve classical algorithms in any way, or lead to more efficient exploration or policy learning? This point is not very clear.

Overall, the paper presents some interesting theoretical results, but there are few practical takeaways for algorithm design, and the proposed similarity metric is intractable to evaluate, which makes approximations to it reduce to pre-existing approaches which are already well-established. Therefore, I am not very convinced of the technical merits of the contributions of this work.

---

> ### Author Response · Authors · 2023-02-03
> **Author response to initial review**
>
> We thank the reviewer for the useful feedback, and are glad to hear their positive comments regarding the theoretical contributions of the paper.
>
> We address the concerns raised below:
>
> (i) Regarding the prior literature of reproducing kernel Hilbert space theory in MDPs: We thank the reviewer for the list of prior work, and we will add a paragraph to our manuscript discussing these. Nevertheless, these methods are quite different from ours, as they all assume a _fixed_ kernel is provided on the state space (either explicitly, or via the assumption of existence of an RKHS). This fixed kernel is then used with standard RL methods such as value iteration, and the resulting algorithm is then analyzed. In contrast, our work _learns_ a kernel on the state space, which classifies states as similar based on their behavioural similarity in the MDP.
>
> (ii) Regarding the comment “…, the reason for adding $d_2$ in the definition of the distance is not very clear, and it additionally cannot be easily evaluated, making it not very useful for algorithmic practical use”: This construction is not new in this work, and has been used in many previous definitions of state metrics, both in theoretical  ([1, 2, 7, 8, 9]) and empirical settings ([3, 4, 5, 6, 10]). The inclusion of the $d_2$ is essential in allowing $d(x, y)$ to go beyond a comparison of the immediate properties of the states $x$ and $y$, such as reward, and instead recursively compare $x$ and $y$ based on the similarity of the states they transition to. We will emphasize this further in the paper. As to the difficulty of evaluation of $d_2$, it depends on the definition of $d_2$ and which assumptions are made. For example, when $d_2$ is the Kantorovich distance, without assumptions it may be difficult to evaluate ([1,2]), however with assumptions such as deterministic ([4]) or Gaussian ([6]) transitions, it becomes simple to evaluate. The $k_2$ term in our kernel definition is the same as the $d_2$ term in the MICo distance ([7]), which can be easily evaluated through independent samples of $(x, a, r, x’)$ pairs.
>
> (iii) Regarding the comments “there are few practical takeaways for algorithm design”, “the proposed similarity metric is intractable to evaluate”: In section 3.5, we demonstrate how our kernel can be learnt in an online fashion using a differentiable loss, and in section 4 we present various large-scale experiments using this loss, so we disagree with the claim of intractability. Regarding practical takeaways for algorithm design, we believe that one of the strongest contributions of the work is providing theoretical justification for the MICo loss introduced in [7], which already provided strong empirical evidence for performance improvements when using these types of losses. We believe that our approach of learning kernels on the state space may be useful for representation learning in reinforcement learning more generally, and that this paper has numerous takeaways, both theoretical and empirical, for algorithm design.

---

> > ### Comment · Reviewer_bNqr · 2023-02-05
> > **Revision**
> >
> > I appreciate the authors addressing my commentary to the best of their ability. However, without uploading the actual revision containing augmented exposition and references, I cannot comment on whether my concerns have been actually addressed. I will await the actual revision's upload before making any determination. At that time, I will use the Compare Revisions feature to study the differences in the manuscripts.

---

> > > ### Author Response · Authors · 2023-02-09
> > > **New revision of manuscript**
> > >
> > > We have just uploaded a new version of the manuscript, taking into account the suggestions from all the reviewers. Although "Compare Revisions" should highlight our changes, we have also used blue text for them for your convenience.

---

### Review · Reviewer_9ozD · 2023-02-04

**Summary Of Contributions:**

The paper addresses the problem of learning embeddings in Reinforcement Learning (RL) by means of Reproducing Kernel Hilbert Spaces (RKHSs). Specifically, the contributions of the paper can be summarized as follows:
- The introduction of a novel distance between value functions based on positive definite kernels defined on the state space. This distance is induced by a fixed-point kernel, which is proven to exist unique tranks to the introduction of a suitable contractive operator.
- The proof that the introduced distance is equivalent to the MICo distance provided in Castro et. al 2021. Furthermore, a novel upper bound on the value function difference is provided, where the bound depends on the introduced distance.
- Several considerations on the ability to embed the introduced distance in the Euclidean space, with bounds on the requested dimensionality to achieve a given accuracy.
- Empirical evaluations, both in evaluation to learn the value function and in control, coupled with deep-RL approaches. These results show some advantages compared to the use of the MICo loss.

**Audience:**

Yes

**Broader Impact Concerns:**

The paper does not contain a "Broader Impact Statement". Nevertheless, the contribution is mainly theoretical and, in the reviewer's opinion, it does not raise ethical concerns.

**Claims And Evidence:**

Yes

**Requested Changes:**

Weaknesses 1, 2, 3, 4, and 5.

**Strengths And Weaknesses:**

***Strengths***
The main point of strength of the paper lies in the theoretical contributions that are significant and novel. The view provided in the paper about the use of RKHSs helps in deepening the understanding of these tools in RL. In particular, the properties of the operator $T^{\pi}_{k}$ (Lemma 8 and 9) are significant.

***Weaknesses***
1. [Abstract] The abstract is slightly too synthetic and, in particular, it does not provide an effective summary of the contributions of the paper. I suggest better contextualizing the contributions.

2. [Introduction] The paragraph of the Introduction devoted to the description of the MICo distance seems too technical for this point of the paper. While it attempts to point out the limitations of the MICo distance, such considerations are likely not appreciated at this point of the paper. I suggest rephrasing the paragraph in a more general way or, possibly, moving it later sections of the paper.

3. [Relation between finite and continuous state space] Most of the results presented in the paper are for *finite* state spaces (e.g., the definition of MDP in Section 2.1, Lemma 9, Corollary 16). However, the authors in Footnote 2 claim that the RKHSs theory holds for "much more general classes of sets". Thus, the authors should clarify which results are limited to finite spaces and which results extend to a continuous one. In my opinion, this is highly relevant since the need for devising an embedding emerges primarily for continuous spaces.

4. [Significance of Theorem 14] Theorem 14 provides a bound to the absolute value function difference into two states ($x$ and $y$) defined in terms of the MICo distance and an additional term which, roughly, depends on the variance of the reward w.r.t. the transition model. This latter term is non-zero even when $x=y$. I see this as a limitation of the bound that clearly shows that it is not tight. Indeed, one expects that the bound should be zero at least when $x=y$. I would appreciate it if the authors could elaborate on this point (here and in the paper) to justify why the bound remains useful even in front of this limitation.

5. [Section 3.5] Section 3.5 seems a little hasty, and the actual rationale behind the loss function in Equation (2) is not properly explained in my opinion. In particular, the meaning of the variables $x'$ and $y'$ is not clear. Do they represent the next states? Furthermore, and more importantly, the properties of the loss function in Equation (2) are not discussed. For instance, is it biased for learning the kernel? I suggest extending this section to provide a more in-depth elaboration and justification behind the choice of the loss function.

***Minor changes***
- In the definition of value function $V^{\pi}$, specify that $R_t$ is sampled from the reward distribution $\mathcal{R}(x_t,a_t)$.
- The definition of $d_{LK}$ should be reported in the main paper.
- The footnote marks should appear after the punctuation symbol.
- The proof of Theorem 13 reported in the main paper references Lemma 21 of Appendix A. Indeed, the actual interesting part of the proof lies in Lemma 21. I suggest either moving the lemma inside the proof in the main paper or postponing the full proof of Theorem 13 to the appendix.
- There is a little notational abuse in the use of $k$ that sometimes denotes the kernel and sometimes the dimension of the embedding.
- Theorem 14: specify that $\mathcal{X}/\sim$ denotes the quotient set w.r.t. the equivalence relation $\sim$.
- Remark 19: I suggest rephrasing it to state that Corollary 16 is improved under the specified condition if we accept an error $\epsilon$.
- Section 4.1: $d_{\sim}^{\pi}$ is not defined in the main paper. Moreover, it does not appear in Figure 1, although it is present in the legend.
- Some equations, especially in the appendix, go beyond the margin.

---

> ### Author Response · Authors · 2023-02-09
> **Response to 9ozD**
>
> We thank the reviewer for their feedback, and are pleased to hear they found our theoretical contributions “significant and novel”.
>
> We address their concerns below:
>
> 1. **[Abstract]** We have expanded our abstract to include more context and a clearer description of our contributions.
> 2. **[Introduction]** We feel having this discussion in the introduction helps to properly contextualize our work. To address your concerns, we have added some additional elaborations to the paragraph in question. Please let us know if you still feel there are points that need further clarification.
> 3. **[Relation between finite and continuous state space]** The intention of Footnote 2 is to alert readers to the fact that the general theory of RKHS applies much more generally, rather than a claim that the original results in this paper apply beyond the finite-state MDP setting. We have reworded the footnote to make this clearer.
> 4. **[Significance of Theorem 14]** The reviewer is correct in the fact that the second term in Theorem 14 can be non-zero even when x = y. This is a consequence of the use of independent couplings in the original MICo metric as well as in our KSMe metric. Indeed, the notion of diffuse metrics (introduced by Castro et el., 2021) precisely captures this “self-distance”. To paraphrase Castro et al., (2021), a non-zero self-distance quantifies the amount of “diffusion” obtained when unrolling trajectories starting from x. The second term in Theorem 14 captures a similar notion. It is worth noting that, in deterministic systems, the second term is in fact zero.
> 5. **[Section 3.5]** We have added further clarifications to this section, and pointers to prior work regarding the properties of the loss presented here.
>
> We have addressed all the other suggested minor changes, thank you for pointing them out.

---

> > ### Comment · Reviewer_9ozD · 2023-03-07
> > **Re: Response to 9ozD**
> >
> > I thank the authors for the feedback and for having revised the paper. Now my concerns have been addressed.

---

### Author Response · Authors · 2023-02-09
**Updated version submitted**

We thank all reviewers for their thoughtful reviews.

We have just uploaded an updated version of the manuscript, taking into account the suggestions from all the reviewers. Although "Compare Revisions" should highlight our changes, we have also used blue text for them for your convenience.

---

### Decision · Action_Editors · 2023-06-06

**Recommendation:** Accept as is

**Comment:**

The updates made by the authors in the last round have successfully addressed the concerns that were raised during the review process. In particular, I appreciate that the authors added further motivation to study the setting they consider, and provided a sufficient amount of technical details in the experimental section, which makes it easier for the reader to appreciate the relevance of the theory to the experiments (and more generally, makes the experimental setup much more clear).

One comment that I will make is that, even after the update, the authors still do not clearly explain why studying distance metrics that *upper-bound* the value differences is of specific interest. In the introduction, they write that "a good embedding should cluster together states which are known to have similar behavioural properties, whilst keeping dissimilar states apart." --- however, the upper-bound property they seek does not eventually guarantee that dissimilar states will have dissimilar values, only that similar states will have similar values. While I don't necessarily expect the authors to share the views I have elaborated on in my earlier comment, it would be useful to admit somewhere in the paper that guaranteeing the upper-bound property does not necessarily provide a complete picture about the usefulness of behavioral metrics, and more research may be needed to provide an understanding about the good empirical performance of the methods studied in this paper and in the related literature.

With that, my recommendation is to accept the paper without major changes. I trust the authors to implement any small changes they deem necessary to respond to the concerns mentioned above.

**Audience:**

Representation learning is a key topic of interest to the TMLR readership. The extensive introduction provided by the paper contextualizes this work adequately in this domain, especially after the updates made in the last round of review. This makes the results reasonably easy to access and understand to the broader TMLR audience.

**Claims And Evidence:**

The technical claims are all supported by rigorous proofs, whose correctness was carefully checked by the reviewers and the action editor.